**Resource**

# Privacy-preserving multicenter differential protein abundance analysis with FedProt

Yuliya Burankova [1,2] ✉, Miriam Abele[1,3], Mohammad Bakhtiari [2], Christine von Toerne [4], Teresa K. Barth[5], Lisa Schweizer [6], Pieter Giesbertz [7], Johannes R. Schmidt [8], Stefan Kalkhof [8,9], Janina Müller-Deile [10], Peter A. van Veelen [11], Yassene Mohammed [11], Elke Hammer [12,13], Lis Arend [2,14], Klaudia Adamowicz [2], Tanja Laske [2,15], Anne Hartebrodt[16,17], Tobias Frisch[16], Chen Meng[3], Julian Matschinske[2], Julian Späth[2], Richard Röttger[16], Veit Schwämmle[18], Stefanie M. Hauck [4], Stefan F. Lichtenthaler [7,19,20], Axel Imhof [5], Matthias Mann [6], Christina Ludwig [3], Bernhard Kuster [1], Jan Baumbach [2,16,21] & Olga Zolotareva [2,14,21]

Quantitative mass spectrometry has revolutionized proteomics by enabling simultaneous quantification of thousands of proteins. Pooling patient-derived data from multiple institutions enhances statistical power but raises serious privacy concerns. Here we introduce FedProt, the first privacy-preserving tool for collaborative differential protein abundance analysis of distributed data, which utilizes federated learning and additive secret sharing. In the absence of a multicenter patient-derived dataset for evaluation, we created two: one at five centers from *E. coli* experiments and one at three centers from human serum. Evaluations using these datasets confirm that FedProt achieves accuracy equivalent to the DEqMS method applied to pooled data, with completely negligible absolute differences no greater than $4 \times 10^{-12}$. By contrast, $-\log_{10}P$ computed by the most accurate meta-analysis methods diverged from the centralized analysis results by up to 25–26.

The expansion of proteomics data is an invaluable resource, unlocking substantial potential for large-scale biomedical research. While genomics provides a static view of an organism's potential capabilities, mass spectrometry (MS)-based proteomics offers detailed insight into dynamic protein composition, interactions and modifications not readily inferred solely from genomics or transcriptomics data[1,2]. The MS-based proteomics enhances our understanding of the proteome's dynamic nature, composition, structure and function.

Techniques such as data-independent acquisition (DIA) MS allow simultaneous quantification of thousands of proteins[3] with wide proteome coverage and low missing values[4,5]. By systematically fragmenting all ions within predefined mass ranges, DIA ensures broad and unbiased peptide identification[6]. This allows novel peptide identification and provides a deep understanding of protein

abundance and posttranslational modifications, crucial in clinical proteomics[7].

In parallel, data-dependent acquisition (DDA), when combined with methods for peptide labeling with tandem mass tags (TMT), has evolved as a versatile clinical proteomics technique[8]. This approach allows simultaneous comparison of peptide abundances across multiple samples in a single MS run. However, it comes with high costs and strict experiment design requirements[9].

Typically, DIA is usually performed without peptide labeling, termed label-free quantification (LFQ). This method is cheaper and requires fewer sample preparation steps, but the accurate quantification of low-abundance proteins is limited[10]. Thus, both label-free DIA and DDA with labeling provide unique strengths and are recognized methods in clinical proteomics.

To maximize clinical proteomics' potential, analyzing larger multicenter patient cohorts is necessary to increase statistical power and achieve more robust results, especially for identifying rare disease subtypes[11,12]. However, integrating patient-derived MS data and proteomics profiles distributed across multiple research institutions can be problematic due to privacy concerns, as they are legally classified as confidential and must be handled accordingly[13]. Similar to transcriptomics, proteomics data can uncover rare sequence variants[14] or be subject to genotype reconstruction attacks[15].

Currently, the only way to collectively analyze distributed proteomic data without compromising patient privacy due to direct data sharing and pooling is to combine individual study outcomes using meta-analysis techniques[16]. Different methodologies present unique advantages and limitations. Meta-analysis performance improves with an increase in sample sizes and number of studies[17] or with the availability of raw data for combined reanalysis[18], which is challenging in proteomics. Common meta-analysis techniques include Fisher's method[16,19], Stouffer's method[16,20], the random effects model (REM)[21,22] and RankProd[23,24].

A consistent limitation of most meta-analyses is that the underlying assumptions about $P$ value or effect size distributions might not hold. In addition, meta-analyses face challenges related to heterogeneity from variations in experimental design, sample characteristics and equipment for peptide separation and MS data acquisition[25]. They cannot fully account for cohort differences, such as target class imbalance or covariate distribution variations[26,27]. Differences in data processing steps, such as normalization, may also substantially impact the results[28].

To enable privacy-preserving analysis of distributed proteomic data owned by multiple institutions while prioritizing data privacy and ensuring robust results despite data heterogeneity, we suggest applying federated learning[29] combined with privacy-enhancing technologies such as secure multiparty computation (SMPC) or additive secret sharing[30]. Recently, the power of a hybrid approach based on federated learning and SMPC to protect privacy during data integration in transcriptomics has been demonstrated by Flimma[27], a privacy-aware tool for differential gene expression analysis of decentralized data. However, Flimma cannot be applied to proteomics data owing to its inability to handle inputs with missing values and the lack of filtering and normalization procedures necessary for MS data. Missing values are intrinsic to proteomics data owing to instrument sensitivity and method design[31] or the stochastic sampling nature of MS, resulting in inconsistent detection of low-abundance proteins[32].

To fulfill an unmet need for a privacy-aware approach tailored for MS-based proteomics, we designed FedProt—a federated learning-based tool for collaborative differential protein abundance analysis of distributed data. FedProt is based on DEqMS, a state-of-the-art limma modification for estimating variance that enhances overall performance[33]. Unlike DEqMS and other tools requiring data centralization, FedProt, by design, preserves patient privacy because the protein abundance profiles always remain in the local environments of the collaborating parties and are never shared externally.

To evaluate FedProt, we used the two most commonly used approaches, LFQ and TMT, and created two multicenter datasets: an LFQ bacterial dataset from five independent centers and a TMT human serum dataset from three. We also used simulated data to test FedProt's behavior under data imbalance. Our results demonstrate that, regardless of data imbalance or batch effects, FedProt always delivers exactly the same results as the original DEqMS workflow.

## Results

### FedProt overview

FedProt represents the mathematical equivalent of DEqMS[33], the accurate variance estimation workflow for MS-based proteomics data. To protect the privacy of patient-derived data, FedProt utilizes the hybrid approach of federated learning[29] and additive secret sharing[30], similar to Flimma[27]. The FedProt workflow overview is shown in Fig. 1.

Federated learning is a machine learning paradigm that increases data privacy by allowing multiple parties to collaboratively train a model without revealing their sensitive data to each other[29]. In this multistep workflow, each participant runs the same application instance (client) that accesses only the participants' local data. Clients compute model parameters based on their local data and exchange them with a central trusted server (coordinator) orchestrating the computations. The coordinator collects local results from clients, aggregates them into global results, and returns them to clients for a new step. The key point is that the local model parameters or intermediate results are constructed to minimize the risk of data reconstruction.

To further enhance privacy and protect local data from reconstruction attacks, we use additive secret sharing[30]. In this method, each client generates multiple noise masks and communicates these masks and masked data to the other parties, ensuring no single party can reconstruct the unmasked data (Fig. 1, right, blue arrows). Each data piece is encrypted with the recipient's public key and summed by receiving parties before being sent to the coordinator. The coordinator receives summarized data and computes and broadcasts the global model to all clients (Fig. 1, black arrows). This scheme allows global aggregation of local results without revealing any local values, thereby enhancing privacy compared with a pure federated learning scheme (see Methods for further details).

To make this decentralized approach and its complex infrastructure available to a broad community, we implemented FedProt as a web-based app with a user-friendly graphical interface. FedProt is published as a certified app in the FeatureCloud[34] app store with documentation and quick-start guidelines (https://featurecloud.ai/app/fedprot). The coordinator sets up a workflow and invites participants. All parties should be registered at https://featurecloud.ai/ and run the FedProt app. While this implementation relies on an internet connection for FeatureCloud coordination, it is fully adaptable. FedProt can be reconfigured to operate on secure, private networks or preapproved communication channels[34], ensuring that all data communications remain within controlled institutional environments.

In the standard configuration, the coordinator configures analysis parameters, for instance, expected number of participants and filtering parameters. Each participant should specify paths to three input.tsv files containing (1) patients' protein intensity profiles, (2) design matrices featuring class labels and covariates and (3) matrices of minimal peptide count across all samples for each unique protein group.

The FedProt federated workflow starts when all invited clients join and provide correctly formatted inputs (Methods). Upon its successful completion, each client receives a table with expression fold changes (FCs), confidence intervals and adjusted $P$ values in the same format as the DEqMS output. The FedProt approach allows us to obtain the same result as centralized pooled data analysis while implementing strong privacy-preserving measures, ensuring no patient-level data are shared and exchanged parameters are hidden from other participants.

### Evaluation approach

Due to privacy regulations and data sharing restrictions, finding publicly available multicenter patient-derived data suitable for evaluating FedProt, given the need for data pooling in centralized analysis to establish the baseline, was challenging. Therefore, specifically for this benchmark, we created two real-world test datasets, one quantified using DIA-LFQ and the other using DDA-TMT (Table 1).

The LFQ-based dataset included 118 *Escherichia coli* colonies cultured under two growth conditions, simulating case and control groups. Of these samples, 98 were unique and uniformly distributed between five independent labs, and four quality control samples were measured by all laboratories (Supplementary Fig. 1).

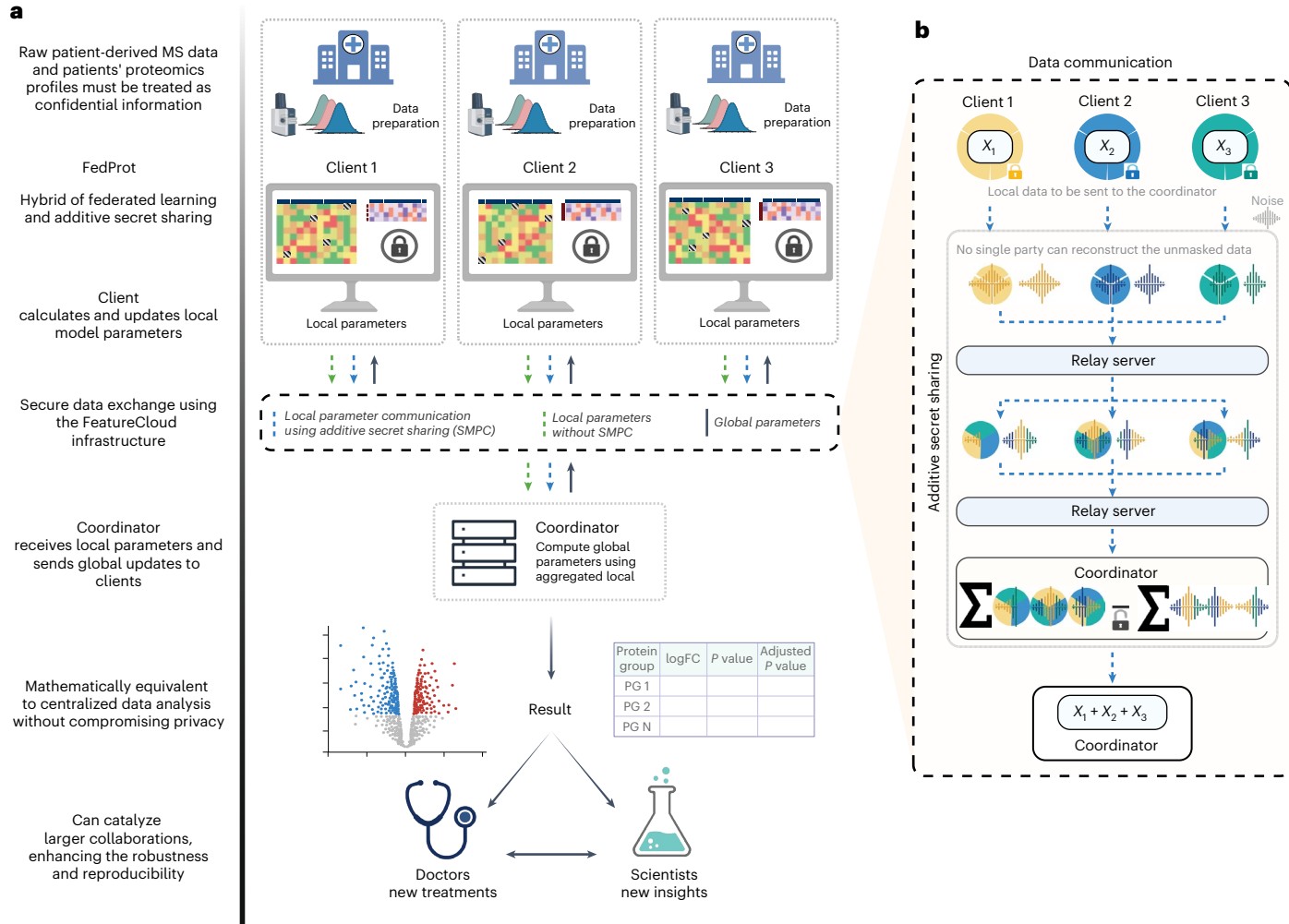

**Fig. 1 | FedProt workflow overview. a**, Federated workflow overview. (1) Data preparation: data owners collect and preprocess MS data, obtain protein intensity and peptide count matrices, and define design matrices before participating as clients. Personal-level information (proteomics profiles) is contained in the client's servers and are never shared (dashed lines around clients highlight different physical locations). (2) Federated learning: clients communicate with the central server (coordinator) to collaboratively train a global model without revealing their individual datasets, but through the exchange of local model parameters. The clients protect their local parameters using additive secret sharing (blue arrows). In case the data are not numeric, such as protein group names, they are sent to the coordinator without additive secret sharing (green arrows). The coordinator returns updated global parameters to clients (black arrows). (3) Result: after all federated computations, all clients receive the results mathematically equivalent to the results of centralized analysis of pooled dataset with DEqMS formatted as a table with abundance FCs, confidence intervals and adjusted *P* values. **b**, Overview of data communication using SMPC (additive secret sharing) inside FedProt. The clients protect their local parameters using additive secret sharing. Each client data point is masked with a noise mask. The noisy data and the noise masks are splitted into *n* encrypted parts (*n* > 2). These parts are exchanged among clients (blue arrows) via a relay server, ensuring that no single party receives more than one piece of the data from each of the other clients. After decrypting the received parts, clients sum the data, and send the reencrypted sums to the coordinator, who decrypts and aggregates the sums to compute the global result. For details, see Methods and Supplementary Methods. Created with BioRender.com.

The TMT-based dataset consisted of three cohorts, each including ten serum samples from individuals with focal segmental glomerulosclerosis (FSGS) and ten control samples. Within each cohort, samples were randomly distributed between two TMT batches, with five samples from each group.

To evaluate FedProt, each center's raw mass spectra were separately quantified using MaxQuant (for the TMT dataset)[35] and DIA-NN (for the LFQ dataset)[36] with the same settings and FASTA files as a database. We assumed that collaborating parties could agree on using a uniform data preprocessing protocol. Although FedProt can tolerate minor variations in preprocessing if a strictly uniform protocol is not feasible, our evaluations on balanced dataset (Supplementary Results) indicate that results from nonuniform preprocessing are similar to those obtained by centralized DEqMS analysis. Nevertheless, we strongly recommend that all parties use the same quantification software for preprocessing whenever possible.

After quantification, pooled data were centrally analyzed using the DEqMS method[33] to establish a baseline (ground truth). We then compared the results of FedProt and four meta-analysis methods: Fisher's[18], Stouffer's[20], the REM[21] and RankProd[23,24] against this baseline. An illustration of the workflows is given in Supplementary Fig. 2.

When analyzing proteomics data from different sources, we encounter incomplete overlap of quantified protein groups between cohorts (Supplementary Fig. 3). The centralized DEqMS method analyzes most features in the aggregated dataset, except those failing the filter on the number of available measurements per target class. Because of privacy concerns, FedProt additionally excludes protein groups with only one measurement per cohort to prevent reconstruction attacks[27]. The R implementations of the Stouffer and RankProd

**Table 1 | Characteristics of datasets for FedProt evaluation with LC–MS/MS measurement overview**

| A. Bacterial dataset | | | | |
|---|---|---|---|---|
| | **Groups**[a] | | **Sample type** | **Setup** |
| | **M9 pyruvate** | **M9 glucose** | | |
| Lab A | 10 | 10 | Cell pellet | Evosep One – Exploris 480 |
| Lab B | 10 | 9 | Cell pellet | nanoElute – timsTOF Pro |
| Lab C | 9 | 10 | Cell lysate | Ultimate3000 – Orbitrap Fusion Lumos |
| Lab D | 10 | 10 | Cell lysate | EASY-nLC 1200 – Exploris 480 |
| Lab E | 10 | 10 | Cell lysate | Ultimate3000 – QE-HFX |
| **B. Human serum dataset** | | | | |
| | **Groups** | | **Sample type** | **Set-up** |
| | **Control** | **FSGS** | | |
| Center 1 | 10 | 10 | Serum | EASY-nLC 1200 – QExactive HF |
| Center 2 | 10 | 10 | Serum | Ultimate3000 nano – Exploris 480 |
| Center 3 | 10 | 10 | Serum | EASY-nLC 1200 – Orbitrap Fusion Lumos |

M9 is the medium used to grow *E. coli*. [a]The number of samples in each cohort in each condition.

**Table 2 | Performance metrics of FedProt and selected meta-analyses using bacterial and human serum datasets**

| Dataset | Method | Mean difference | Maximal difference | FP[a] | FN[a] | Jaccard similarity coefficient[a] |
|---|---|---|---|---|---|---|
| Bacterial dataset | FedProt | **$4.45 \times 10^{-13}$** | **$3.56 \times 10^{-12}$** | **0** | **0** | **1** |
| | Fisher | 3.86 | 25.24 | 0 | 1 | 0.998 |
| | Stouffer | 3.35 | 25.00 | 0 | 1 | 0.998 |
| | REM | 15.61 | 259.80 | 8 | 14 | 0.964 |
| | RankProd | 13.42 | 79.42 | 0 | 108 | 0.820 |
| Human serum dataset | FedProt | **$1.36 \times 10^{-13}$** | **$6.59 \times 10^{-13}$** | **0** | **0** | **1** |
| | Fisher | 0.50 | 2.79 | 2 | 6 | 0.922 |
| | Stouffer | 0.57 | 2.59 | 2 | 13 | 0.854 |
| | REM | 0.59 | 11.64 | 1 | 13 | 0.863 |
| | RankProd | 1.07 | 8.98 | 1 | 33 | 0.667 |

Mean and maximum absolute differences in negative log-transformed BH-adjusted *P* values, Jaccard similarity coefficients, and error rates (FPs and FNs) for the results of FedProt and selected meta-analysis approaches compared with centralized DEqMS results. The lowest differences are shown in bold font. [a]Thresholds of |log$_2$FC| >0.5 and adjusted *P* value <0.05 (bacterial dataset) and |log$_2$FC| >0.25 and adjusted *P* value <0.05 (human serum dataset) were used for Jaccard similarity coefficients, FPs and FNs. Metrics use each method's own |log$_2$FC| and BH-adjusted *P* values.

methods can aggregate the results only for protein groups presenting in all cohorts; Fisher's method requires presence in at least two cohorts. The REM can use all input protein groups, although, as we will observe in the later analysis, this does not improve the quality of its results.

Supplementary Fig. 4 illustrates this limitation and quantifies the features lost due to decentralization for the bacterial and human serum datasets. We evaluated the results of differential abundance analysis considering only the features that all the methods could process.

In addition, to investigate the effect of data imbalance on the results of decentralized methods, we created simulated data with increasing levels of imbalance across cohorts (Methods).

### Deviations in the results of decentralized methods

FedProt produced results that matched the results of the centralized DEqMS workflow in all tests for both datasets. This was evident in the consistency between the mean absolute difference for adjusted *P* values and logFC values, as the maximum absolute differences were negligible (no greater than $4 \times 10^{-12}$; Table 2 and Supplementary Table 1).

By contrast, the meta-analysis showed notable deviations from the ground truth, with mean differences in log-transformed *P* values ranging from 3.35 to 15.61 for the LFQ dataset and 0.50 to 1.1 for the TMT dataset (Table 2 and Fig. 2a). Similarly, the logFC results from the

meta-analyses demonstrated larger differences, with maximal absolute differences up to 0.2 (Supplementary Table 1 and Supplementary Fig. 5). To compare logFCs, we used the results of REM and Fisher's method since Fisher's method calculates it using the same approach as Stouffer's method and RankProd.

Furthermore, we addressed batch effects, which can confound results if not properly managed. For both datasets, FedProt showed perfect correlation in logFCs with DEqMS results applied to batch-corrected data, despite differences in handling batch effects (see Supplementary Results for more details).

This superior consistency of FedProt and centralized DEqMS results proves that the federated approach can achieve the same results as the centralized model but with the considerable advantage of privacy protection.

### The consistency of differentially abundant protein lists

Averaged absolute differences quantify the discrepancy between the centralized and decentralized methods for all protein groups, given that the errors computed for both differentially and nondifferentially abundant proteins are treated equally during statistical analysis (Table 2 and Supplementary Table 1). However, in many studies, the exact *P* values and effect sizes are not as crucial as accurate identification and

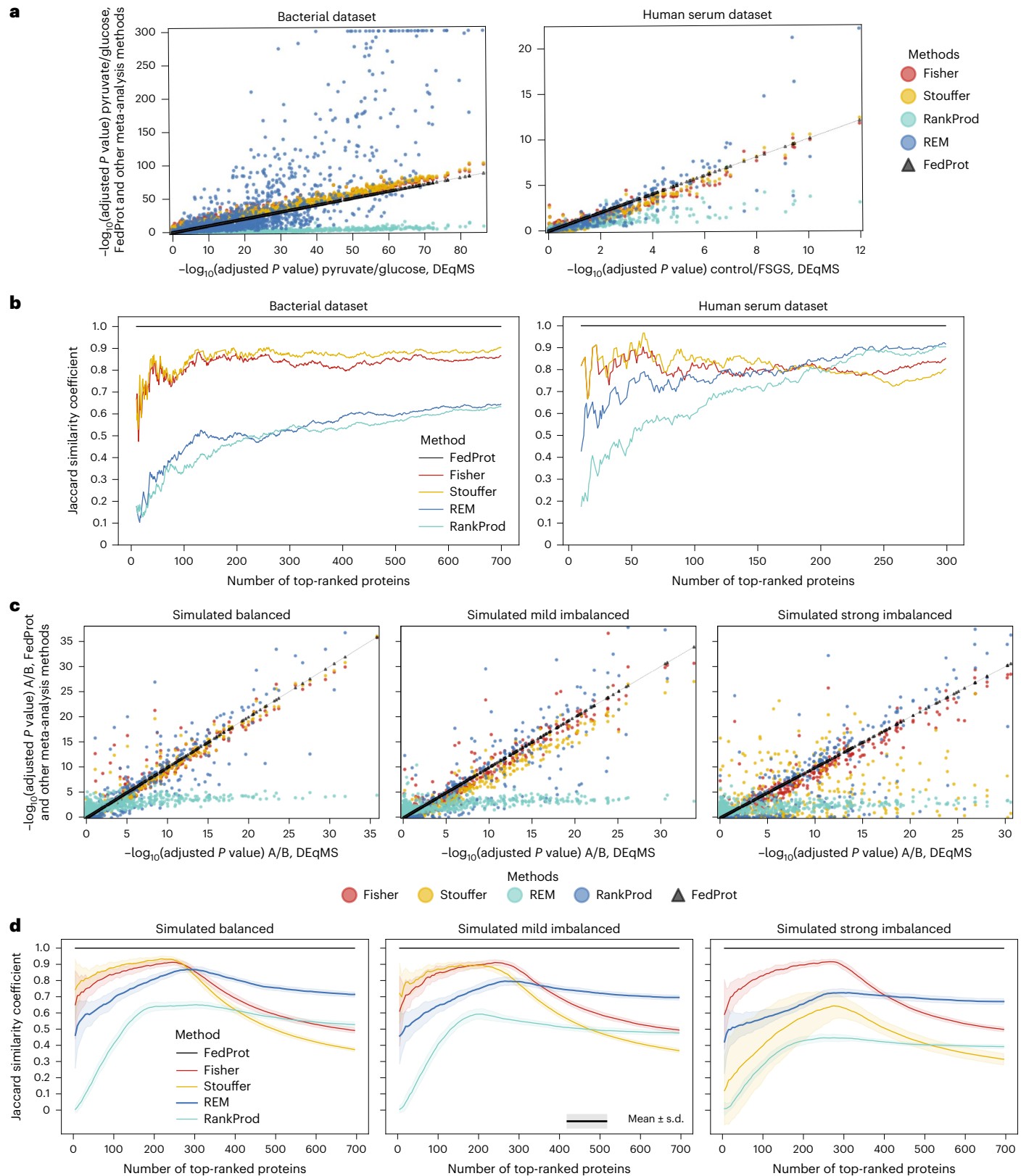

**Fig. 2 | Comparative analysis of adjusted P values and ranking consistency between centralized and decentralized methods for real and simulated datasets. a,c**, The comparison of negative log-transformed BH-method adjusted P values ($-\log_{10}$(adjusted P value)) computed by FedProt or meta-analysis methods (y axis) with the centralized DEqMS analysis (x axis), for bacterial and human serum datasets (**a**) and for simulated datasets (**c**), for one out of the 50 runs per scenario. The thin black line is the diagonal. **b,d**, The dependency of the Jaccard similarity coefficient on the number of top-ranked proteins identified by the centralized DEqMS and decentralized approaches, showing the results for the bacterial and human serum datasets (**b**) and for the simulated datasets (**d**). Proteins were ranked on the basis of their decreasing negative log-transformed BH-adjusted P values and not filtered by $\log_2$FC. The simulated data generation and the subsequent analysis were repeated 50 times, with aggregated results reported (mean values ± s.d.).

**Table 3 | Performance metrics of FedProt and selected meta-analyses using simulated datasets**

| Dataset | Method | Mean difference | Maximal difference | FP | FN | Jaccard similarity coefficient |
|---|---|---|---|---|---|---|
| Balanced | FedProt | **$3.22 \times 10^{-15} \pm 3.52 \times 10^{-16}$** | **$1.85 \times 10^{-13} \pm 8.71 \times 10^{-14}$** | **0.0±0** | **0.0±0** | **1.00±0.00** |
| | Fisher | 0.12±0.01 | 12.50±4.14 | 2.3±1.5 | 2.2±1.4 | 0.95±0.02 |
| | Stouffer | 0.14±0.01 | 8.99±3.39 | 2.3±1.5 | 2.6±1.4 | 0.94±0.02 |
| | REM | 0.15±0.01 | 18.90±4.79 | 6.2±2.7 | 7.4±2.5 | 0.84±0.04 |
| | RankProd | 0.78±0.02 | 27.60±4.00 | 11.1±3.4 | 1.0±0.9 | 0.87±0.03 |
| Mild imbalance | FedProt | **$6.00 \times 10^{-15} \pm 6.22 \times 10^{-16}$** | **$2.67 \times 10^{-13} \pm 7.30 \times 10^{-14}$** | **0.0±0** | **0.0±0** | **1.00±0.00** |
| | Fisher | 0.14±0.01 | 13.70±4.65 | 18.2±3.9 | 17.1±4.6 | 0.64±0.04 |
| | Stouffer | 0.20±0.01 | 8.70±2.18 | 17.8±3.9 | 17.6±4.5 | 0.64±0.04 |
| | REM | 0.19±0.01 | 23.50±4.06 | 8.1±2.6 | 10.3±3.8 | 0.79±0.04 |
| | RankProd | 0.84±0.02 | 32.10±3.93 | 38.9±6.7 | 16.0±4.3 | 0.54±0.04 |
| Strong imbalance | FedProt | **$1.33 \times 10^{-14} \pm 1.69 \times 10^{-15}$** | **$6.63 \times 10^{-13} \pm 2.74 \times 10^{-13}$** | **0.0±0** | **0.0±0** | **1.00±0.00** |
| | Fisher | 0.15±0.01 | 13.20±4.67 | 40.0±5.9 | 29.4±4.7 | 0.45±0.04 |
| | Stouffer | 0.41±0.06 | 29.20±8.11 | 33.4±4.1 | 36.3±5.6 | 0.41±0.05 |
| | REM | 0.22±0.01 | 26.20±4.94 | 7.8±2.8 | 15.3±3.3 | 0.75±0.04 |
| | RankProd | 0.91±0.02 | 34.40±5.62 | 147.0±11.8 | 28.2±4.8 | 0.25±0.02 |

Mean and maximum absolute differences in negative log-transformed BH-adjusted P values, Jaccard similarity coefficients and error rates (FPs and FNs) for the results of FedProt and selected meta-analysis approaches compared with centralized DEqMS results. The values corresponding to the best performance between all methods are highlighted in bold font. Mean and standard deviation are reported for $n=50$ of simulation and the subsequent data analysis runs. Jaccard similarity coefficients and error rates were computed with $|\log_2 FC|>1$ and adjusted P value <0.05 thresholds. Metrics use each method's own $|\log_2 FC|$ and BH-adjusted P values.

consistent ranking of differentially abundant proteins. Therefore, we compared the lists of the most strongly and significantly differentially abundant protein groups detected by decentralized methods.

As before, we compared these lists with the ground truth from the centralized DEqMS method. We applied $|\log_2 FC| > 0.5$ and adjusted P value of 0.05 thresholds for the bacterial dataset, and for the human serum dataset 0.25 and 0.05, respectively. The $|\log_2 FC|$ thresholds were selected on the basis of their distributions in the datasets. Method performances in terms of false positives (FPs), false negatives (FNs) and Jaccard similarities are presented in Table 2. In addition, we evaluated a range of $|\log_2 FC|$ and adjusted P value thresholds to examine how different cutoffs affect FedProt and meta-analysis methods' performance (Supplementary Results).

Out of all tested methods, only FedProt identified exactly the same differentially abundant protein groups as centralized DEqMS. Regardless of the method used, the outputs of all meta-analysis methods always contained FPs and FNs, with the deviation from ground truth being higher in the human serum dataset despite its smaller number of protein groups. FedProt maintained robust performance even under varying thresholds, although the meta-analyses showed greater discrepancies, especially for smaller $|\log_2 FC|$ cutoffs, as for the human serum dataset (Supplementary Figs. 6–8).

In addition, by ranking proteins on the basis of the adjusted P values, we assessed how method performance shifted with changing the number of top differentially abundant protein groups (Fig. 2b). Identifying a limited number of significantly differentially abundant protein groups is often a key objective in studies such as biomarker discovery. Like in previous tests, FedProt consistently matched the results of the centralized approach in both test scenarios, outperforming all meta-analysis methods.

### Robustness against data imbalance

We further tested FedProt's results reliability even when faced with the challenge of data imbalance using simulated data. We generated intensity matrices for three scenarios (balanced, mildly imbalanced, and strongly imbalanced data), each with 6,000 proteins and 600 samples from two conditions using the same approach as described in ref. 37 (Supplementary Table 2). Batch effects were introduced

using the ComBat model[38], and missing values were added per ref. 39. In addition, we simulated a confounder in condition B, with varying frequencies across cohorts. As our focus was to investigate the effect of data imbalance on the results, the analyses were concluded without count adjustment. Each simulation and analysis was repeated 50 times.

Regardless of data imbalance, FedProt produced results that closely matched those from the centralized DEqMS workflow in all scenarios, with exceptionally small differences. The maximum absolute difference of adjusted P values and logFC values did not exceed $6.7 \times 10^{-13}$ (Table 3 and Supplementary Table 3).

As the degree of data imbalance increased, the maximum and mean absolute differences for meta-analyses consistently increased (Table 3, Fig. 2c and Supplementary Table 3). Data imbalance substantially affected not only P values obtained by meta-analyses but also logFCs (Supplementary Table 3 and Supplementary Fig. 9). By contrast, FedProt stably demonstrated robust, error-free performance.

Furthermore, we analyzed how the data imbalance affects the list of differentially abundant proteins using $|\log_2 FC| > 1$ and Benjamini–Hochberg (BH)-adjusted P value of 0.05 thresholds. Regardless of the method used, all meta-analysis outputs contained FPs and FNs, and their counts grew with an increasing degree of data imbalance (Table 3). By contrast, FedProt's results remained stable under varying degrees of imbalance.

Regarding the dependence of the Jaccard similarity index on the number of top-ranked protein groups, again, meta-analyses showed higher discrepancy at the top of the list; it grew with the increase of data imbalance (Fig. 2d). At the same time, FedProt's results completely match the results of the central analysis, regardless of dataset or imbalance.

In addition, we evaluated FedProt's scalability using simulated datasets with 66, 660 and 6600 samples (Supplementary Results). The results confirmed that FedProt consistently replicates centralized analysis results across varying cohort sizes, whereas meta-analyses, even with larger sample sizes, do not fully reproduce the central results, especially under data imbalance. These findings further highlight the robustness and scalability of FedProt over standard meta-analysis approaches.

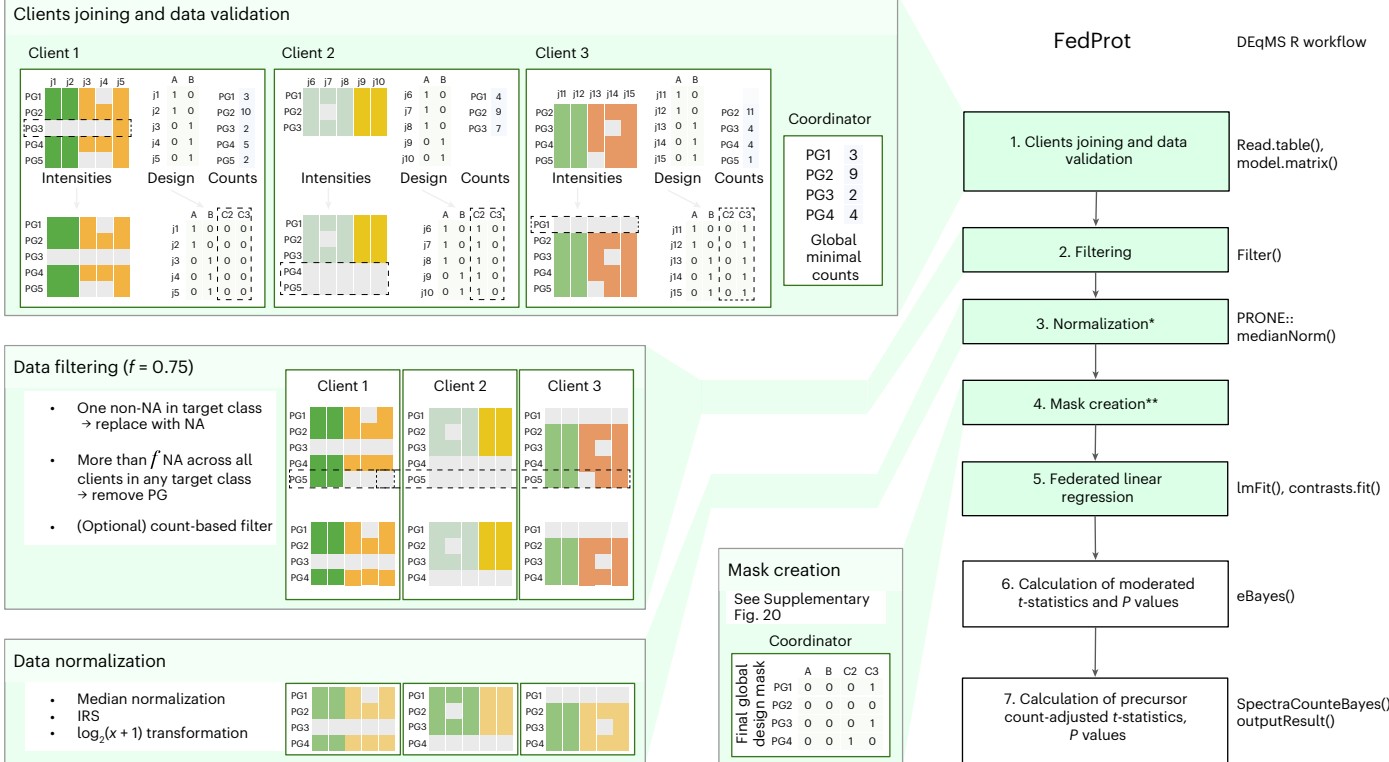

**Fig. 3 | Scheme of the FedProt workflow.** Steps that involve federated computations are shown in green. The corresponding stages of DEqMS workflow are shown on the right. Median normalization from the PRONE[58] R package was used. The validation, filtering, normalization and design mask creation steps involving three clients (C1, C2 and C3) are shown on the left. PG denotes protein groups, j are sample numbers, and A and B are target classes compared during the analysis. On step 1, the PG3 value for sample j5 is replaced with NA (only one not-missing value in the client data). On step 2, the PG5 value for sample j5 is replaced with NA (one not-missing value for this PG in the target class for this client). After that, the whole PG5 group is removed from all clients because of too few nonmissing values (here, less than $f$ = 0.75). On the design mask creation step, client 3 for PG1 is excluded because it has no data, same for PG4 of client 2; and, the client 3 became the new reference client and also excluded from computation (PG3 is missing in client 1). See Methods and Supplementary Methods for more details. *The normalization step is optional and could be turned off by the coordinator. In case of TMT data, for filtering out decoys, contaminant and reverse protein groups are required. The normalization by median across all centers and IRS inside each center can be performed during a FedProt run. **The step is for the federated approach only. Created with BioRender.com.

## Robustness against preprocessing variability

FedProt effectively integrates heterogeneous protein quantification data from multiple centers despite variations in liquid chromatography–mass spectrometry (LC–MS) setups, software versions and preprocessing protocols, achieving performance equivalent to centralized DEqMS analysis while outperforming meta-analysis approaches when raw data reanalysis is unfeasible. Using a bacterial dataset processed with different quantification software (Supplementary Table 4), FedProt delivered results comparable to centralized analysis (Supplementary Results and Supplementary Fig. 10).

Furthermore, integration of publicly available proteomics datasets for clear-cell renal cell carcinoma[40–42] (Supplementary Table 5) using gene names as the common identifier further confirmed FedProt's high consistency with centralized DEqMS, as indicated by negligible differences in log$_2$FC and negative log-transformed adjusted $P$ values (Supplementary Results). This evaluation demonstrates FedProt's capability of handling nonuniform preprocessing and its potential utility for multicenter studies, particularly in scenarios where access to raw data is limited.

## Discussion

In this study, we introduce FedProt—a privacy-preserving tool for federated differential protein abundance analysis. The results from FedProt and centralized DEqMS analysis are nearly identical because the federated approach uses the same linear modeling and variance estimation steps as centralized DEqMS, with additive secret sharing enabling the secure aggregation of local intermediate statistics. The minor differences observed are attributable solely to floating-point arithmetic variations between R (limma/DEqMS) and Python (FedProt).

However, our study is not devoid of limitations. First, the current version of FedProt supports only LFQ and TMT proteomics data and two normalization methods. However, FedProt's design is inherently adaptable, and the current workflow can be concluded without count adjustment. It allows future extensions to other data types such as phosphoproteomics[43] or metabolomics[44], and to multi-omics data analysis[45].

Second, the federated learning approach enables privacy-preserving analysis of distributed data but cannot guarantee absolute privacy alone. By using additive secret sharing[30,34], FedProt enhances privacy protection compared with pure federated learning, ensuring the local parameters' original values remain hidden from the central server. In addition, it includes built-in checks and alerts for client-side data anomalies, which will stop the computation if the number of clients involved exceeds the total number of samples. Besides, such scenarios are very unlikely, and the likelihood of reconstruction attacks is extremely low[46,47].

Overall, FedProt provides enhanced privacy protection compared with traditional centralized analysis at the cost of negligibly small errors compared with errors of meta-analyses. It is a promising approach with the potential to facilitate larger-scale, privacy-preserving multicenter collaborations in clinical proteomics.

## Methods

### FedProt workflow

FedProt is based on the accurate variance estimation workflow of DEqMS[33] for MS data and uses the hybrid approach of federated learning and additive secret sharing[30] in a manner similar to Flimma[27]. Federated learning is a machine learning paradigm in which models are trained on multiple devices (clients) without centralizing data, enabling collaborative learning with increased data privacy. Clients securely store and analyze their local data and exchange intermediate results with the trusted server (Coordinator) that aggregates local results to global.

Before the federated analysis, each participant preprocesses its local dataset independently and is responsible for ensuring the quality and consistency of the data provided to the client app. We assume that all the participants used the same protocol to quantify and preprocess their local data.

The FedProt's federated analysis workflow is divided into six (or seven if normalization is needed) steps, of which four (or five) involve federated computations and require one or several rounds of communication between client and server (Fig. 3).

In the first step, $k > 2$ FedProt clients join the analysis initialized by the coordinator and validate their data. From each client $c^i$, where $i = 1, \dots, k$, FedProt requires two tab-separated (.tsv) files:

(1) A protein intensities matrix $Y^i_{raw}$ containing the intensities of $n^i$ proteins or protein groups (rows) detected in $m^i$ samples (columns). Protein groups detected only in a single sample in the cohort are replaced with NA (missing value), to avoid exchanging individual-level data in the next steps and protect data privacy.

(2) A design matrix $X^i$ specifying which columns of the intensity matrix belong to which experimental conditions or groups. In the design matrix, each experimental condition or group should be coded as a binary variable, with 1 if the sample belongs to the group and 0 if not. For example, if we compare conditions A and B, the design should contain both A and B columns with 0 or 1 for each sample. Optionally, the design matrix can contain columns representing the covariates. The names of target class columns and covariates in local design matrices must match the names specified by the coordinator during the initialization of the study.

(3) An optional file—a matrix with the number of quantified precursor peptides for each protein group $Pr^i_{min}$. For each client, one value per protein group is required. For example, it can be found in the Precursor.Ids column in the DIA-NN report, or in the Peptide.IDs column in the MaxQuant report. If the clients do not have $Pr^i_{min}$, the computation will be completed after the sixth step, with no precursor peptide count adjustment.

Clients join the server, and during the first step FedProt ensures that all the clients provide all necessary inputs. Each client sends to the server information about the number of samples $m^i$, the peptide-to-protein minimal counts $Pr^i_{min}$ and the list of protein groups $P^i$ they have. The server uses each client's $Pr^i_{min}$ matrices to get the global minimal number of quantified precursor peptides across all samples from all clients for each protein group, $Pr^i_{min}$; this is needed for the last step.

For each client $c^i$, the server updates the set of all detected protein groups $P$ across all clients participating in the analysis. After that, clients receive from the server the set of all protein groups $P$, and the list of variables $v$ accounted for in the model (target classes, list of cohorts and, optionally, covariates). If the protein group is not in the client data, it is created and filled with NAs in the $Y^i_{raw}$.

The list of variables $v$ is used to update the design matrix $X^i$ and include columns representing batches to account for batch effects. Each client updates the design matrix $X^i$ and adds cohort effects to it

based on the list of variables $v$. The cohort effects are added as binary columns, where the first client $c^1$ represents the reference batch, as in limma, and the corresponding column is not included in the design matrix. If the coordinator participates in the computation as a client, it becomes the first client $c^1$.

In the second step, FedProt applies filters. One of them is to filter out protein groups with too many missing values. Each client $c^i$ calculates the number of samples with missing values per target class for each protein group and shares it with the server. The server computes global fractions of missing values per protein group and target class and the protein groups not detected in more than $f$ fraction of each target class samples in any target classes are removed from $P$. The value of $f$ is set to 0.8 by default and can be adjusted by the coordinator. The next optional filter removes protein groups supported by only one peptide precursor; this filter can be enabled by the coordinator. Another filter converts rows to missing values (NA) if all but one value are missing in samples for a given design column; this protects sensitive data sharing during computations and is done before the first above-described filter.

The next step is the normalization step; this step is optional and depends on the coordinator settings. Currently, two types of normalization are implemented. The first one is median normalization. For that, each client calculates the median intensity across all protein groups for each sample $j$ in their dataset, $Med^i_j$. Clients' median average $\overline{Med}$ is sent to the coordinator. The coordinator then calculates the global weighted mean of clients' sample medians:

$$\overline{Med} = \frac{\sum_{i=1}^{k} \overline{Med}^i \times m_i}{\sum_{i=1}^{k} m^i}.$$

Once the global median mean is computed, the coordinator broadcasts it to clients. Each client's $j$th sample intensity values are adjusted on the basis of this value:

$$Y^i_{norm,j} = \frac{Y^i_{raw,j}}{Med^i_j} \times \overline{Med}.$$

The second normalization is internal reference scaling (IRS) using in silico references. This normalization is suitable when one client has multiple TMT plexes and is conducted within each cohort. For each TMT plex, an in silico reference sample $Ref_{plex}$ is created taking the mean value for each protein group across all samples in the TMT plex. Then, the geometric mean for each protein across all clients' in silico references is computed as

$$GM = \left( \exp \frac{1}{d} \sum_{1}^{d} \log \left( Ref_{plex} \right) \right),$$

where $d$ is the total number of plexes in the $i$th client.

Using $Ref_{plex}$, the scaling factor $SF_{plex}$ is calculated for each protein in each TMT plex as the ratio of the geometric mean to the in silico reference for that plex:

$$SF_{plex} = \frac{GM}{Ref_{plex}},$$

and normalized intensities are computed by multiplying with the scaling factor.

Both implemented normalization methods should be done on non-log-transformed data, so after this step, $\log_2(x + 1)$ log transformation is applied if required by the coordinator in the analysis settings.

To make possible the analysis of all protein groups available, we used a design matrix mask $D$ for the next steps. The mask has the number of columns equal to the design matrix and rows for each protein group. The mask creation is described in more detail in Supplementary Methods.

In the fifth step, for each protein group in $P$, FedProt fits a linear model in a federated fashion, following the approach described by Karr et al.[48], assuming that protein group intensity $Y$ can be modeled as

$$Y = X\beta + \epsilon,$$

where $X$ is the global design matrix and $\epsilon$ is random noise. The coefficients $\beta$ defining the impact of each variable in the design matrix $X$ on the observed intensities can be estimated as

$$\hat{\beta} = (X^T X)^{-1} X^T Y,$$

and the unscaled standard deviations st.dev$_{unscaled}$ or the coefficients $\hat{\beta}$ can be estimated as

$$\text{st.dev}_{unscaled} = \sqrt{\text{diag}\left((X^T X)^{-1}\right)}.$$

To avoid sharing $X^i$ and $Y^i$ containing sensitive patient-level data that would be necessary to obtain $X$ and $Y$, $X^T X$ and $X^T Y$ terms of the equation can be computed through the summation of local $(X^i)^T X^i$, and $(X^i)^T Y^i$ computed by clients:

$$X^T X = \sum_{i=1}^{k} (X^i)^T X^i$$

$$X^T Y = \sum_{i=1}^{k} (X^i)^T Y^i,$$

$(X^i)^T X^i$ and $(X^i)^T Y^i$ do not reveal any patient-level data and can be shared with the server. To minimize the rare risk of data exposure, clients independently detect and resolve issues during the data validation step (protein groups detected in only one cohort sample are replaced with NA; Fig. 3, step 1).

On this step design mask $D$ is used to exclude columns and rows corresponding to missing values from $X^T X$ and columns from $X^T Y$. This is necessary to exclude from the calculations for a particular protein the values belonging to a particular column from the design $X$, because these cohorts do not have values (all are NAs). $D$ usage allows us to simulate behavior of the lmFit function from the limma R package when working with missing values in data.

To minimize the risk of reconstruction attack, $(X^i)^T X^i$, $(X^i)^T Y^i$ and any local computation result shared with the aggregating server are protected by additive secret sharing[30]. In brief, each client generates $n$ randomly sampled noise masks, $r_1, \ldots, r_n$, as equally distributed random values and calculates corresponding noisy data as $(M - r_1 - \ldots - r_n)$. This noisy data, alongside the noise masks (divided into $n$ pieces in total), is communicated with other computational parties via a relay server, ensuring that no party receives more than one piece (one data piece and one noise piece) from any specific client. The data pieces are encrypted with each party's public key to ensure the data cannot be intercepted. Once the encrypted pieces are received, each party decrypts and sums the received data, then sends the reencrypted received sums to the coordinator again via the relay server. The coordinator gets the summed parts and, in case of sending $(X^i)^T X^i$ and $(X^i)^T Y^i$, obtains the global coefficients $\hat{\beta}$.

During the additive secure aggregation process, the noises cancel out due to their additive nature, resulting in the correct global aggregation of local parameters without any noticeable impact on the final outcome compared with nonsecure aggregation. This cancelation mechanism preserves data privacy, as the individual intermediate results that remain are not revealed to the coordinator or any other parties. By increasing the number of parties, the risk of collusion to reconstruct the clients' intermediate results will be further reduced. To simplify the technical aspects of communicating data, FeatureCloud[34] passes all data through the relay server that cannot decrypt the data (see details in Supplementary Methods).

Global estimated coefficients $\hat{\beta}$ are shared with the clients, who use it to calculate the local sum of squared errors SSE$^i = \sum_{j}^{m_i}(y_j^i - \hat{y}_j^i)^2$, where $\hat{y}_j^i$ is the $j$th component of $\hat{Y}^i = X^i \hat{\beta}$. The server aggregates local sum of squared errors SSE$^i$ to the global sum of squared errors SSE,

$$SSE = \sum_{i=1}^{k} SSE^i$$

and computes the residual standard deviations $\sigma$ as follows:

$$\sigma = \sqrt{\frac{SSE}{m - |v|}},$$

where $m = \sum_{i}^{k} m^i$ is the number of samples with detected protein group and $|v|$ is the number of variables in the design matrix except masked with the design matrix mask. In this step, a design mask is applied to ensure that missing values are handled correctly.

The sixth step is performed solely on the server side. Given a set of target classes, the contrast is defined as linear combinations of conditions or target classes in a design matrix $X$ and represented as a contrast matrix $K$. The fitting of these contrasts implies the application of the contrast matrix to the regression coefficients $\hat{\beta}$ and covariance matrix of these coefficients $C$ as follows:

$$\hat{\beta}' = \hat{\beta} K$$

$$C' = K^T C K.$$

The standard deviations st.dev for the coefficients are also updated during this step. Specifically, the covariance matrix $C'$ is scaled by its diagonal to become the correlation matrix, on which the Cholesky decomposition is performed, and the result is then used to transform and scale the st.dev and get the st.dev$'$. This replicates the implementation of the contrasts.fit function from the limma R package[49].

Further computations performed on the server side replicate eBayes from limma[50] and require only global $\hat{\beta}'$, $\sigma^2$ and st.dev$'$. This step starts with moderated $t$-statistic calculation. For that, variance shrinkage is performed to stabilize the variance estimates across genes. Given a vector of sample variances $\sigma^2$ and their associated degrees of freedom df$_{residual}$, the empirical Bayes approach fits an $F$ distribution to estimate the parameters of the prior distribution.

Using the estimated priors (variance $\sigma_{prior}^2$ and the degrees of freedom df$_{prior}$), the posterior variances $\sigma_{post}^2$ were calculated as a weighted average of the prior and sample variances as follows:

$$\sigma_{post}^2 = \frac{df_{residual}\sigma^2 + df_{prior}\sigma_{prior}^2}{df_{residual} + df_{prior}}.$$

After this, the moderated $t$-statistic $t$ and $B$-statistic $B$ are computed as

$$t = \frac{\beta'}{\text{st.dev}'} \times \frac{1}{\sqrt{\sigma_{post}^2}},$$

where $\beta'$ represents the estimated coefficients (logFCs) and st.dev$'$ denotes the unscaled standard deviations of the coefficients, and

$$B = \log\left(\frac{p}{1-p}\right) - \frac{\log(r)}{2} + k,$$

where $p$ is the proportion of differentially expressed genes, $r$ is the ratio of the variance of the gene to the prior variance and $k$ is a function

involving the moderated *t*-statistics and the degrees of freedom. The *B*-statistic represents the log-odds of differential expression.

Finally, the BH procedure[51] is applied to compute adjusted *P* values. In the result of this step, a feature table that provides moderated *t*-statistics, logFCs, confidence intervals and adjusted *P* values is generated. The FedProt workflow can be completed after this step without precursor peptide count adjustment.

The last, seventh step replicates spectraCounteBayes from the DEqMS method[33], which estimates different prior variances for proteins quantified by different numbers of peptide precursors and calculates peptide count-adjusted moderated *t*-statistics and *P* values. The server uses the minimal number of quantified precursor peptides across all samples for each protein for estimating the variance of log-counts and fitting a regression model. As a result, the system outputs a final table with statistical measurements for each feature corrected to the number of precursors.

The resulting table is saved on a server and sent to the clients. This approach in FedProt allows us to obtain the same result as what centralized pooled data analysis would yield while implementing strong privacy-preserving measures.

The FedProt user-friendly implementation is accessible as a FeatureCloud app (https://featurecloud.ai/app/fedprot), making privacy-preserving differential protein abundance analysis available to a broad community of biomedical experts. In addition, the FeatureCloud platform supports the integration of multiple applications into workflows. This allows FedProt to be combined with other privacy-preserving analysis tasks (for example, federated singular value decomposition[52] and clustering techniques such as *k*-means clustering[53] available as FeatureCloud apps).

## Meta-analysis approaches

To evaluate FedProt's accuracy in comparison with meta-analyses, we used three classes of meta-analyses: effect size combination methods (the REM[54,55] from the metaVolcanoR package[56] v.1.12.0), *P* value combination methods (Fisher's method[19] from the metaVolcanoR package[56] v.1.12.0 and Stouffer's method[20] from the the MetaDE[55] package v.2.2.3) and nonparametric rank combination methods[54,57] (Rank Product method from the RankProd[24] R package v.3.24.0).

For all chosen meta-analysis methods except REM, the global FC was calculated as the mean of local FCs, producing the same values. Consequently, only Fisher's method and REM results were utilized for the evaluation of logFCs.

## Human serum data

**Sample preparation.** For FedProt evaluation, we were using a TMT dataset of 60 independent human blood serum samples, comprising 30 from patients with primary FSGS and 30 from healthy controls. Written consent for anonymized data retrieval and storage was obtained. The local ethics committee of the Friedrich-Alexander Universität Erlangen–Nürnberg provided approval for the nephrological biobank of the Klinikum Bayreuth (ethics approval code 264_20 B) and the proteomics analysis (ethics approval code 221_20 B). Approval to perform MS of serum samples was given by the ethics committee of the Friedrich-Alexander Universität Erlangen–Nürnberg (ethics approval code 182_19 B).

The samples were separated into three groups, each containing ten healthy and ten FSGS samples, blinded and distributed to three studies centers by the clinical partners (Supplementary Methods). The samples were prepared and measured by independent researchers (that is, in three different LC–MS/MS locations on different days) in a blinded manner using the same protocol until complete data collection. Patient group allocations were disclosed for data analysis.

**Sample preparation and LC–MS/MS measurement.** Samples were prepared by three independent scientists applying a harmonized

protocol. TMT-labeled samples were combined into six sets, each containing five healthy, five FSGS and one common reference sample.

MS data were acquired in three independent research centers using their preferred instruments and corresponding parameter setups (Table 1B). More details on the LC–MS/MS measurement protocols are provided in Supplementary Methods.

**Raw data analysis.** Raw mass spectra were uniformly preprocessed and quantified using MaxQuant (v.2.4.2) software[35] separately for each center. The analysis was conducted with default settings unless otherwise specified. Experimentally acquired mass spectra were searched against a human reference proteome (Uniprot, v.2023_05, reviewed/Swiss-Prot entries only, 20,418 protein sequences). Trypsin/P was set as protease (specific mode) allowing a maximum of two missed cleavages. Carbamidomethylation of cysteine was set as a fixed modification, while oxidation of methionine and acetylation of protein N-terminus were allowed as variable modifications. A minimum of two peptides including one unique peptide were required for protein inference controlling the false discovery rate to <0.05 in target/decoy mode. Match between runs (MBR) was enabled.

**Data filtering and preprocessing.** For protein intensity matrices, MaxQuant reports were independently preprocessed filtering out decoy, contaminant and modification site-only entries. The column 'Majority. protein.IDs' was used for protein group names, and within a group, proteins were sorted alphabetically. This additional sorting allows a better intersection of independently processed data. As FedProt uses in silico references in the TMT analysis workflow, six reference samples were removed from the dataset before analysis.

Protein groups supported by a single peptide were removed during the central DEqMS analysis. Raw intensities were normalized to the median across all data (from PRONE[58] R package v.1.0.4, https://github.com/daisybio/PRONE), followed by IRS within each center with an in silico reference—the mean of all samples for each pool in the center (modified IRS from PRONE). Similar filters and normalizations are also implemented in FedProt. Quality control was performed in R (Supplementary Figs. 1 and 10B).

## Bacterial dataset creation

**Sample preparation.** We evaluated FedProt using a LFQ dataset of 118 samples generated from *E. coli* MG1655 (DSM 18039) cultures. Samples were shipped on dry ice either as lysates or as cell pellets (Table 1A; for more details, see Supplementary Methods).

Laboratory A and laboratory B received cell pellets, while others (C, D and E) received already lysed cells. Each laboratory received 20 (19 for laboratories B and C) unique and 4 shared (quality control) samples, 12 (11) samples per condition. One sample was lost during shipment, and one more was excluded during quality control of MS results. Each of the four quality control samples was generated by aliquoting one starting sample, meaning they are technical replicates.

**Sample preparation and LC–MS/MS measurement.** Laboratories A, B and C performed bacteria cell lysis separately, and laboratories E and D used cell lysates prepared by laboratory C, sent and diluted. Each laboratory used slightly different protocols for protein digestion, peptide purification and preparation for MS.

MS data were acquired in five independent research centers using their preferred sample preparation (in case of cell pellets), MS measurement protocols, instruments and corresponding parameter setups (Table 1A). More details on the LC–MS/MS measurement protocols are provided in Supplementary Methods.

**Raw data analysis.** Raw mass spectra were uniformly preprocessed and quantified using DIA-NN[36] (https://github.com/vdemichev/DIA-NN), v.1.8.1 in a separate run for each laboratory. The analysis was conducted

with default settings unless otherwise specified. When analyzing the robustness against preprocessing variability, Spectronaut v. 17.5 and v.17.2 and DIA-NN v.1.8.0 and v.1.8.1 were used (Supplementary Table 4). An in silico spectral library was generated in each DIA-NN and Spectronaut run from the *E. coli* MG1655 (taxID 83333) protein sequence database (Uniprot UP000000625, 4,448 entries) provided via a FASTA file (no preexisting spectral library was utilized).

The analysis was carried out in 'any LC (high-accuracy)' mode. Two missed cleavages and a maximum of two variable modifications per peptide were allowed: acetylation of protein N-termini and oxidation of methionine. Minimum precursor $m/z$ was set to 360. MBR was enabled. The data were reanalyzed utilizing a deep-learning-generated spectral library to refine the results. For specific parameters on the setup of the DIA-NN searches, see Supplementary Table 6.

**Data filtering and preprocessing.** Protein quantities were obtained from the MaxLFQ[59] algorithm implemented in DIA-NN v 1.8.1 and extracted from reports using the diann v.1.0.1 R package. For protein intensity matrices, DIA-NN outputs were filtered using the following criteria: Lib.Q.Value ≤0.01 and Lib.PG.Q.Value ≤0.01. As MaxLFQ protein quantities are already normalized, no additional normalization was executed during preprocessing or implemented in FedProt.

Quality control was performed in R (Supplementary Figs. 1 and 11), one sample from laboratory C's dataset was excluded after quality control due to being an outlier (Supplementary Fig. 12).

## Simulated data
To generate simulated data, we used the approach proposed in ref. 37:

$$y_{pj} \approx \begin{cases} \mathcal{N}\left(\mu_p, \sigma_p^2\right) & \text{w.p. } (1 - \pi_p) \\ \mathcal{N}\left(\mu_p + \Delta\mu, \sigma_p^2\right) & \text{w.p. } \pi_p \end{cases},$$

where $y_{pj}$ represents the intensity for $p$th protein, $p = 1, \ldots, n$, from $j$th sample, $j = 1, \ldots, m$.

Protein intensity $y_{pj}$ is modeled from mixture distribution, where $\pi_p \in [0, 0.5)$ is the outlier proportion. Outliers could be differentially abundant proteins or technical errors. We did not add a sample effect to our model because we simulated data after the MaxLFQ method[59], which contains the normalization step eliminating it. The protein population distribution parameters were generated separately for each protein, means $\mu_p$ were from $N(0, 2)$ and variances $\sigma_p^2$ were from an inverse gamma distribution.

We adapted the sim.dat.fn function from the RobNorm package[37] v.0.2.0, with modifications to the inverse gamma distribution parameters (the shape parameter of 2 and scale parameter of 3). Parameter $\Delta\mu$ was used to represent a shift in a differentially regulated block, to generate up- or downregulated proteins. The protein in the block has a chance, derived from a binomial distribution with a success probability of 0.8, of undergoing a shift. $\Delta\mu$ was used to represent a shift in a differentially regulated block, to generate up- or downregulated proteins.

We generated the data without batch effects first, with 6,000 proteins and 600 samples, 300 each in conditions A and B. The block consisting of 200 proteins differentially abundant between conditions A and B was obtained using $\Delta\mu = 1.25$. To simulate the effects of unknown covariate, a block of 150 proteins each was generated, with $\Delta\mu = 1.25$, and randomly assigned to samples in class B. The proportion of samples in class B with unknown covariate is presented in Supplementary Table 2.

To simulate a multicenter study, we then randomly split the data into three cohorts (C1, C2 and C3) and added batch effects. To investigate the effect of data imbalance on method performances, data were split into cohorts twice: once with equal cohort sizes and frequencies of conditions A and B, and once with unequal cohort sizes and condition frequencies (see Supplementary Table 2 for details).

To simulate batch effects in our data, we utilized the ComBat model[38] designed for removing batch effects:

$$y_{pji} = y_{pj} + \gamma_{pi} + \delta_{pi}\epsilon_{pji},$$

where $i = 1, \ldots, k$, and $k$ is the total number of batches, errors $\epsilon_{pji}$ are normally distributed $N(0, 1)$, additive batch parameter $\gamma_{pi}$ is drawn from normal distribution and multiplicative batch parameter $\delta_{pi}$ is drawn from inverse gamma distribution. For simulation, we used $\gamma_{p1} \approx N(0, 1)$ for additive batch effects and $\delta_{p1} \approx IG(3, 2)$ for multiplicative batch effect for the first batch, $\gamma_{p2} \approx N(0.2, 0.5)$ and $\delta_{p2} \approx IG(2.5, 1)$ for the second batch, and $\gamma_{p3} \approx N(-0.2, 1.5)$ and $\delta_{p3} \approx IG(4, 0.5)$ for the third batch.

Missing values were introduced to the data using the approach described previously[39], with a missing value rate of 0.2 and a missing-not-at-random rate of 0.5, resulting in a total of 20% missing values in the dataset.

## Data analysis
Data after quantification and preprocessing were analyzed in R v.4.2.0. Batch effects correction and plots were done using the following R packages: limma v.3.54.2, data.table v.1.14.8, gridExtra v.2.3, patchwork v.1.1.2, reshape2 v.1.4.4, matrixStats v.1.3.0 and tidyverse v.2.0.0 (includes ggplot2 v.3.4.2, dplyr v.1.1.4, purrr v.1.0.2, readr v.2.1.4 and tidyr v.1.3.1).

Filtering based on the number of missing values per class was done using the 80% threshold, the same as the FedProt default value. For bacterial datasets, the transformation of rows with only one non-NA value per subset of samples for a design column to NA is disabled for evaluation. For the TMT dataset, median normalization from the PRONE[58] package v.1.0.4 was used (https://github.com/daisybio/PRONE).

Differential protein abundance in centralized analysis was tested with a two-sided moderated $t$-test (with empirical Bayes and with BH adjustment for $P$ values) using DEqMS v.1.16.0 (R 4.2.0). We used log$_2$FC, exact BH-adjusted $P$ values and full output tables for the bacterial and human serum datasets (provided in Supplementary Table 7). FedProt's adjusted $P$ values used for evaluation using the bacterial and human serum datasets are count-scaled BH-adjusted $P$ values, similar to what DEqMS calculates. Before log transformation of adjusted $P$ values, a small value ($1 \times 10^{-300}$) was used to replace 0 in REM results for the bacterial dataset. For evaluation using the simulated dataset, BH-adjusted $P$ values were not scaled using counts, because we did not simulate spectra counts data.

FedProt implementation on the FeatureCloud platform was evaluated using Python library FeatureCloud v.0.0.32, with rpy2 v.3.5.11 Python library inside the app Docker container (Docker v.27.1.2). For the meta-analyses, we used MetaDE[55] v.2.2.3, metaVolcanoR[56] v.1.12.0 and RankProd[24] v.3.24.0 R packages. The results of DEqMS runs in each center or laboratory separately were used as input data for meta-analysis methods.

Evaluation was done using Python using pandas v.2.2.2, numpy v.2.0.0, scipy v.1.14.0, scikit-learn v.1.5.0 and statsmodels v.0.14.2 libraries. For upset plots, Python upsetplot library v.0.9.0 was used (https://upsetplot.readthedocs.io/en/stable/). For other plots, matplotlib v.3.8.4 and seaborn v.0.13.2 packages were used. SMPC was enabled during the evaluation.

To investigate the impact of including batch effects in the design, the data were preprocessed in the same way, depending on the dataset. The difference lay in incorporating batch information into the design during the analysis or correcting the data beforehand (after preprocessing) and then performing the analysis without including this information in the design.

## Reporting summary
Further information on research design is available in the Nature Portfolio Reporting Summary linked to this article.

## Data availability

The MS proteomics data have been deposited to the ProteomeXchange Consortium via the PRIDE[60] partner repository with the dataset identifiers PXD053812 (the bacterial dataset, https://www.ebi.ac.uk/pride/archive/projects/PXD053812) and PXD053560 (the human serum dataset, https://www.ebi.ac.uk/pride/archive/projects/PXD053560). Uniprot human reference proteome version 2023_05 (Uniprot UP000005640, reviewed/Swiss-Prot entries only, 20,418 protein sequences, https://www.uniprot.org/proteomes/UP000005640) and *Escherichia coli* (strain K12) reference proteome (Uniprot UP000000625, 4448 entries, https://www.uniprot.org/proteomes/UP000000625) were used for the human serum and bacterial datasets, respectively. Three datasets (clear-cell renal cell carcinoma) were used for studying robustness against preprocessing variability from the PRIDE[60] repository (https://www.ebi.ac.uk/pride/) with the dataset identifiers PDC000127 ref. 41, PXD042844 ref. 42 and PXD030344 ref. 40. Data to obtain them are available via Zenodo at https://doi.org/10.5281/zenodo.15370419 (ref. 61) or at the FedProt repository via GitHub at https://github.com/Freddsle/FedProt (inside the data and evaluation folders). The example datasets (bacterial, human serum and three simulated scenarios used in the Resource) can be found in the same repository inside the data folder. Source data are provided with this paper.

## Code availability

The code used for the data preprocessing, quality control and the simulated dataset generation, together with the code for running and evaluating FedProt, is available via GitHub at https://github.com/Freddsle/FedProt, via FeatureCloud App Store at https://featurecloud.ai/app/fedprot or via Zenodo at https://doi.org/10.5281/zenodo.15370419 (ref. 61).

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

## Acknowledgements

We acknowledge F. Gruhn, A. Preuss and U. Scholz for technical assistance. This work was supported by the German Federal Ministry of Education and Research (BMBF) within the framework of CLINSPECT-M/-2 (grant numbers 161L0214A, 03LW0243K, 03LW0244 and 03LW0248) to Y.B., M.M., S.F.L, K.T.B., A.I., C.L., B.K., J.B., S.M.H. and O.Z. This work was supported by the German Federal Ministry of Education and Research (BMBF) within the framework of Sys_CARE (grant numbers 01ZX1908D and 01ZX2208D) to E.H. This work was supported by the German Federal Ministry of Education and Research (BMBF) within the framework of the e:Med research and funding concept, project SyMBoD (grant numbers 01ZX2210B, 01ZX1910B, 01ZX1910D and 01ZX2210D) to J.B., T.L., J.R.S. and S.K. This project has received funding from the European Union's Horizon2020 research and innovation program under grant agreement number 826078 (J.B., M.B., A.H., T.F., J.M., J.S. and R.R.). This publication reflects only the authors' view, and the European Commission is not responsible for any use that may be made of the information it contains. This work was developed as part of the FeMAI project and is funded by the German Federal Ministry of Education and Research (BMBF) under grant number 01IS21079 to K.A. and J.B. This work was funded by the German Federal Ministry of Education and Research (BMBF) (grant number 16DTM100A) to J.B. This work was funded by the German Federal Ministry of Education and Research (BMBF) (grant number 01GM2202D) to J.M.-D. This work was funded by the Deutsche Forschungsgemeinschaft (DFG, German Research Foundation) under Germany's Excellence Strategy within the framework of the Munich Cluster for Systems Neurology (EXC 2145 SyNergy—ID 390857198) to S.F.L. This work was supported by the Federal Ministry of Education and Research (BMBF) and the Universität Hamburg with funds of the Excellence Strategy of the Federal Government and the Länder (to

L.A. and T.L.). This work was supported by the German Federal Ministry of Education and Research (BMBF, grant number 031L0309A) to A.H. This research was supported by CVDLINK project (M.B. and J.B.). This project was awarded funding by the European Union's Horizon Research and Innovation program under grant agreement number 101137278. Views and opinions expressed are, however, those of the author(s) only and do not necessarily reflect those of the European Union or the European Health and Digital Executive Agency. Neither the European Union nor the granting authority can be held responsible for them.

## Author contributions

Y.B., M.B., C.M., V.S. and O.Z.—FedProt differential abundance analysis algorithm development. M.B., A.H., T.F., J.M., J.S. and R.R.—FedProt privacy-preserving measures development. Y.B., M.B. and O.Z.—FedProt implementation and testing. M.A.—bacterial dataset generation. Y.B., M.A., C.v.T., T.K.B., L.S., P.G., S.M.H., S.F.L., A.I., M.M., C.L. and B.K.—bacterial dataset measurement, quantification and preprocessing. J.R.S., S.K., J.M.-D., P.A.v.V. and Y.M.—human serum dataset generation, measurement and quantification. Y.B., L.A. and T.L.—human serum dataset preprocessing. Y.B., L.A. and O.Z.—simulated data generation. Y.B., E.H., L.A., K.A., T.L. and O.Z.—data analysis and interpretation. Y.B., M.A., M.B., C.v.T., T.K.B., L.S., P.G., J.R.S., C.L., J.B. and O.Z.—drafted the paper and designed the figures (with input from all authors). J.R.S., S.K., C.L., B.K., J.B. and O.Z.—designed the study and supervised the work. All authors have read and approved the final version of the paper.

## Funding

## Competing interests

B.K. is a co-founder and shareholder of MSAID; he holds no operational role in the company. The other authors declare no competing interests.

## Additional information

**Correspondence and requests for materials** should be addressed to Yuliya Burankova.

[1]Chair of Proteomics and Bioanalytics, TUM School of Life Sciences, Technical University of Munich, Freising, Germany. [2]Institute for Computational Systems Biology, University of Hamburg, Hamburg, Germany. [3]Bavarian Center for Biomolecular Mass Spectrometry, TUM School of Life Sciences, Technical University of Munich, Freising, Germany. [4]Metabolomics and Proteomics Core, Helmholtz Center Munich, Munich, Germany. [5]Protein Analysis Unit, Biomedical Center, Faculty of Medicine, LMU Munich, Martinsried, Germany. [6]Max Planck Institute of Biochemistry, Martinsried, Germany. [7]German Center for Neurodegenerative Diseases (DZNE), Munich, Germany. [8]Department of Preclinical Development and Validation, Fraunhofer Institute for Cell Therapy and Immunology IZI, Leipzig, Germany. [9]Institute for Bioanalysis, University of Applied Science Coburg, Coburg, Germany. [10]Department of Nephrology, Uniklinikum Erlangen, Friedrich-Alexander-Universität Erlangen-Nürnberg, Erlangen, Germany. [11]Center for Proteomics and Metabolomics, Leiden University Medical Center, Leiden, the Netherlands. [12]University Medicine Greifswald, Greifswald, Germany. [13]German Center for Cardiovascular Diseases (DZHK), Greifswald, Germany. [14]Data Science in Systems Biology, TUM School of Life Sciences, Technical University of Munich, Freising, Germany. [15]Viral Systems Modeling, Leibniz Institute of Virology, Hamburg, Germany. [16]Department of Mathematics and Computer Science, University of Southern Denmark, Odense, Denmark. [17]Department of Artificial Intelligence in Biomedical Engineering, Friedrich-Alexander-Universität Erlangen–Nürnberg, Erlangen, Germany. [18]Department of Biochemistry and Molecular Biology, University of Southern Denmark, Odense, Denmark. [19]Neuroproteomics, School of Medicine and Health, Klinikum rechts der Isar, Technical University of Munich, Munich, Germany. [20]Munich Cluster for Systems Neurology (SyNergy), Munich, Germany. [21]These authors jointly supervised this work: Jan Baumbach, Olga Zolotareva. ✉e-mail: yuliya.burankova@tum.de

# Reporting Summary

## Statistics

For all statistical analyses, confirm that the following items are present in the figure legend, table legend, main text, or Methods section.

| n/a | Confirmed | |
|---|---|---|
| ☐ | ☒ | The exact sample size (*n*) for each experimental group/condition, given as a discrete number and unit of measurement |
| ☐ | ☒ | A statement on whether measurements were taken from distinct samples or whether the same sample was measured repeatedly |
| ☐ | ☒ | The statistical test(s) used AND whether they are one- or two-sided *Only common tests should be described solely by name; describe more complex techniques in the Methods section.* |
| ☐ | ☒ | A description of all covariates tested |
| ☐ | ☒ | A description of any assumptions or corrections, such as tests of normality and adjustment for multiple comparisons |
| ☐ | ☒ | A full description of the statistical parameters including central tendency (e.g. means) or other basic estimates (e.g. regression coefficient) AND variation (e.g. standard deviation) or associated estimates of uncertainty (e.g. confidence intervals) |
| ☐ | ☒ | For null hypothesis testing, the test statistic (e.g. *F*, *t*, *r*) with confidence intervals, effect sizes, degrees of freedom and *P* value noted *Give P values as exact values whenever suitable.* |
| ☒ | ☐ | For Bayesian analysis, information on the choice of priors and Markov chain Monte Carlo settings |
| ☒ | ☐ | For hierarchical and complex designs, identification of the appropriate level for tests and full reporting of outcomes |
| ☒ | ☐ | Estimates of effect sizes (e.g. Cohen's *d*, Pearson's *r*), indicating how they were calculated |

*Our web collection on statistics for biologists contains articles on many of the points above.*

## Software and code

Policy information about availability of computer code

| Data collection | The raw mass-spectrometry data were collected using the following combinations equipment and the software supplied with it and  Evosep One – Exploris 480; nanoElute – timsTOF Pro; Ultimate3000 – Orbitrap Fusion Lumos; EASY-nLC 1200 – Exploris 480; Ultimate3000 – QE-HFX; EASY-nLC 1200 – QExactive HF; Ultimate3000 nano – Exploris 480; EASY-nLC 1200 – Orbitrap Fusion Lumos. |
|---|---|
| Data analysis | MaxQuant v 2.4.2, DIA-NN v 1.8.1 and v.1.8.0, Spectronaut v 17.5 and v. 17.2, Docker v.27.1.2. Python (v.3.11.9) packages:  pandas v.2.2.2, numpy v.2.0.0, statsmodels v.0.14.2, scipy v.1.14.0, matplotlib v.3.8.4, seaborn v.0.13.2, scikit-learn v.1.5.0, upsetplot v.0.9.0, plotly v.5.22.0. Python package featurecloud v.0.0.32 and related packages to FeatureCloud app run: rpy2 v.3.5.11. R (v.4.2.0) libraries: DEqMS v1.16.0, limma v3.54.2, diann v1.0.1, RobNorm v0.1.0, invgamma v1.1, RankProd v3.24.0, MetaVolcanoR v1.12.0, metaDE v2.2.3, ggrepel v0.9.3, data.table v1.14.8, gridExtra v2.3, patchwork v1.1.2, reshape2 v1.4.4, matrixStats v1.3.0, tidyverse v2.0.0 (includes ggplot2 v3.4.2, dplyr v1.1.4, purrr v1.0.2, readr v2.1.4, tidyr v1.3.1), PRONE v1.0.4. Code for running the analysis is located in GitHub repo: https://github.com/Freddsle/FedProt/tree/main. |

For manuscripts utilizing custom algorithms or software that are central to the research but not yet described in published literature, software must be made available to editors and reviewers. We strongly encourage code deposition in a community repository (e.g. GitHub). See the Nature Portfolio guidelines for submitting code & software for further information.

## Data

Policy information about availability of data

All manuscripts must include a data availability statement. This statement should provide the following information, where applicable:
- Accession codes, unique identifiers, or web links for publicly available datasets
- A description of any restrictions on data availability
- For clinical datasets or third party data, please ensure that the statement adheres to our policy

The mass spectrometry proteomics data have been deposited to the ProteomeXchange Consortium via the PRIDE partner repository with the dataset identifiers PXD053812 (the bacterial dataset, https://www.ebi.ac.uk/pride/archive/projects/PXD053812) and PXD053560 (the human serum dataset, https://www.ebi.ac.uk/pride/archive/projects/PXD053560). Uniprot human reference proteome version 2023_05 (Uniprot UP000005640, reviewed/Swiss-Prot entries only, 20,418 protein sequences, https://www.uniprot.org/proteomes/UP000005640) and Escherichia coli (strain K12) reference proteome (Uniprot UP000000625, 4448 entries, https://www.uniprot.org/proteomes/UP000000625) were used for the human serum and bacterial datasets, respectively. Three datasets (clear-cell renal cell carcinoma, ccRCC) were used for studying robustness against preprocessing variability from the PRIDE66 repository (https://www.ebi.ac.uk/pride/) with the dataset identifiers: PDC000127, PXD042844, PXD030344.
Results and code to obtain them are available via Zenodo at https://doi.org/10.5281/zenodo.15370419 or at the FedProt repository via GitHub at https://github.com/Freddsle/FedProt (inside the data and evaluation folders). The minimal datasets (bacterial, human serum, and three simulated scenarios used in the main manuscript) can be found in the same repository inside the data folder. Source data for Figure 2 is available with this manuscript.

## Human research participants

Policy information about studies involving human research participants and Sex and Gender in Research.

| | |
|---|---|
| Reporting on sex and gender | Both male (n=31) and female (n=29) samples were included, age 19-79 (Supplementary table 12). The gender was based on self-reporting. We have written consent to share the data. The gender was not included in the study design. No sex or gender analysis have been included. |
| Population characteristics | 60 independent human blood serum samples, comprising 30 from patients with primary FSGS and 30 from healthy controls. Clinical characteristics (control or FSGS) and covariate characteristics are reported in Supplementary Table 12. |
| Recruitment | Samples were taken from the nephrological biobank of the Klinikum Bayreuth. Written consent for anonymized data retrieval and storage was obtained. |
| Ethics oversight | The local ethics committee of the Friedrich-Alexander Universität Erlangen-Nürnberg. |

Note that full information on the approval of the study protocol must also be provided in the manuscript.

# Field-specific reporting

Please select the one below that is the best fit for your research. If you are not sure, read the appropriate sections before making your selection.

☒ Life sciences  ☐ Behavioural & social sciences  ☐ Ecological, evolutionary & environmental sciences

For a reference copy of the document with all sections, see nature.com/documents/nr-reporting-summary-flat.pdf

# Life sciences study design

All studies must disclose on these points even when the disclosure is negative.

| | |
|---|---|
| Sample size | No formal statistical power analysis was performed as our objective was to compare our method to a central reference, not to discover novel effects. Sample sizes were selected based on data availability and on prior experience with similar datasets. This study used a TMT human plasma dataset (60 samples: 2 conditions, 3 cohorts), an LFQ Escherichia coli dataset (118 samples: 2 conditions, 5 cohorts), and simulated data (600 samples: 2 conditions, 3 cohorts). These sample sizes reflect those commonly used in biomarker discovery studies, ensuring adequate representation of the expected variability. |
| Data exclusions | 1 outlier was excluded in E.coli MS-based dataset after data quality control. As our workflow relied on in silico reference samples, 6 pooled reference samples measured in TMT human serum dataset were not used in the analysis. |
| Replication | We tested our method on a variety of real-word and simulated datasets to ensure reproducibility and robustness of the results. Analysis performed independently on real datasets 3 times (3 multi-center datasets used) and 150 times for simulated datasets (50 runs for each scenario). |
| Randomization | E.coli samples were randomly distributed across 5 research centers. For synthetic data, each data simulation and analysis was repeated 50. times. Human serum dataset samples were blinded and distributed to three studies centers by the clinical partners. |
| Blinding | Human serum dataset samples were blinded and distributed to three studies centers by the clinical partners. Samples were prepared and |

| Blinding | measured by independent researchers in a blinded manner until complete data collection. Patient group allocations were disclosed for data analysis. Revealing the information about samples, e.g. class labels, could not influence on the result of the experiments. |

# Reporting for specific materials, systems and methods

We require information from authors about some types of materials, experimental systems and methods used in many studies. Here, indicate whether each material, system or method listed is relevant to your study. If you are not sure if a list item applies to your research, read the appropriate section before selecting a response.

## Materials & experimental systems

| n/a | Involved in the study |
|-----|----------------------|
| ☒ ☐ | Antibodies |
| ☒ ☐ | Eukaryotic cell lines |
| ☒ ☐ | Palaeontology and archaeology |
| ☒ ☐ | Animals and other organisms |
| ☒ ☐ | Clinical data |
| ☒ ☐ | Dual use research of concern |

## Methods

| n/a | Involved in the study |
|-----|----------------------|
| ☒ ☐ | ChIP-seq |
| ☒ ☐ | Flow cytometry |
| ☒ ☐ | MRI-based neuroimaging |

