## [Peer Review File · Nature Computational Science]

Privacy-Preserving Multi-center Differential Protein Abundance Analysis with FedProt

Corresponding Author: Ms Yuliya Burankova

Version 0:

Decision Letter:

** Please ensure you delete the link to your author homepage in this e-mail if you wish to forward it to your co-authors. **

Dear Ms Burankova,

Your manuscript "Privacy-Preserving Multi-center Differential Protein Abundance Analysis with FedProt" has now been seen by 3 referees, whose comments are appended below. You will see that while they find your work of interest, they have raised points that need to be addressed before we can make a decision on publication.

The referees' reports seem to be quite clear. Naturally, we will need you to address **all** of the points raised.

While we ask you to address all of the points raised, the following points need to be substantially worked on:

- Most systems used for the handling of sensitive data do not have access to the internet, or the connection is very restricted. Please address how this is handled.
- Please discuss how the performance scales with the number of samples.
- Please describe "additive secret sharing."
- The sentence, "Due to privacy regulations, finding publicly available multicenter patient-derived data suitable for evaluating FedProt was challenging," is somewhat ambiguous. It gives the impression that FedProt might not comply with privacy regulations. This should be clarified.
- Please explore other proteomics analysis task that could be performed under privacy constraints.
- To increase the usability, it would be beneficial to assess the performance of FedProt with data processed using different preprocessing protocols.
- Please demonstrate the performance of FedProt in integrating quantification tables from different software.
- Since statistical significance is based on the adjusted p-value, it would be interesting to investigate the results using different cutoffs for log₂ fold-change.

Please use the following link to submit your revised manuscript and a point-by-point response to the referees' comments (which should be in a separate document to any cover letter):

Link Redacted

** This url links to your confidential homepage and associated information about manuscripts you may have submitted or be reviewing for us. If you wish to forward this e-mail to co-authors, please delete this link to your homepage first. **

To aid in the review process, we would appreciate it if you could also provide a copy of your manuscript files that indicates your revisions by making use of Track Changes or similar mark-up tools. Please also ensure that all correspondence is marked with your Nature Computational Science reference number in the subject line.

In addition, please make sure to upload a Word Document or LaTeX version of your text, to assist us in the editorial stage.

To improve transparency in authorship, we request that all authors identified as 'corresponding author' on published papers create and link their Open Researcher and Contributor Identifier (ORCID) with their account on the Manuscript Tracking

System (MTS), prior to acceptance. ORCID helps the scientific community achieve unambiguous attribution of all scholarly contributions. You can create and link your ORCID from the home page of the MTS by clicking on 'Modify my Springer Nature account'. For more information please visit www.springernature.com/orcid.

We hope to receive your revised paper within three weeks. If you cannot send it within this time, please let us know.

Best regards,

Ananya Rastogi, PhD
Senior Editor
Nature Computational Science

Reviewers comments:

Reviewer #1 (Remarks to the Author):

In this manuscript, Burankova et al. present a solution for federated quantitative proteomic analysis. The problem of jointly analyzing proteomic data is increasing as proteomics becomes available for cohorts. Overall, the work is well executed and the paper well written. I have mainly suggestions regarding points of discussion but otherwise recommend the publication of this paper.

Main comments

- Most systems used for the handling of sensitive data do not have access to the internet, or the connection is very restricted. How is this handled by the author's approach? What are the requirements of the proposed approach in terms of software environment? How would their system work on a setup like the one of the UK biobank?
- As the authors noticed, getting access to data is hard. In fact, experience says that in meta-analyses coordination of analysts is more of a problem than data processing. And if the protocol is too much of a burden for the analyst, the meta-analysis will simply never happen. Can the authors comment on the ease of their method in comparison to a standard meta-analysis?
- The sample sizes proposed by the authors for testing are very small, modern cohort proteomic data feature several hundreds or thousands of samples. How does the performance scale with the number of samples? Will the gain of running a federated analysis over a standard meta-analysis still be significant?

Minor writing details

- In the introduction, avoid considerations on 'comprehensiveness'. Also avoid ranking methods or having judgment on which is best - this is not the right place for such considerations.
- When considering data privacy, please avoid mixing legal considerations with potential threats. Rare variant identification, reidentification, reconstruction are not the reason why privacy should be preserved. The data are private, by law, they should be handled as such. If it would be a standalone birth year, albeit totally harmless these data would still need to be handled in a privacy-preserving system.
- Please provide basic summary statistics on the result of the proteomic quantitative analyses.
- In Figure 1, please clearly highlight that the personal-level information is contained in the local servers and are never shared. This could also be made clearer in the text.
- Please define what 'masks' refer to in the context of the paper.

Reviewer #1 (Remarks on code availability):

I have inspected the code but did not try to install or run the application.

The repository is very well documented. The code is clear and structured.

Reviewer #2 (Remarks to the Author):

In this manuscript, Burankova and colleagues describe a federated learning approach for protein abundance analysis. Federated learning is a critical methodology for working with sensitive patient data, as it enables data analysis without the need to pool sensitive information at a single site. For this, the authors propose the use of federated learning with multi-party computation. This approach mirrors a method previously proposed by the same authors for analyzing gene expression data. A novel aspect of this work is its consideration of proteomics-specific challenges, such as the measurement of distinct proteins and platform-specific artifacts.

Overall, this is a relevant and sound approach. The authors conducted evaluation experiments of data from two distinct platforms (DDA and DIA); and simulated data for robustness analysis regarding batch artefacts and class imbalance.

Results indicate that FedProt achieves comparable or identical performance to the non-federated version and that it outperforms competing methods, which are primarily based on meta-analysis. The manuscript is well written, but a few recommendations could further improve its quality.

Specific points

Results for FedProt (Fig. 2 and Table 3) appear very similar to the non-federated DEqMS. Could this be due to the fact that federated learning algorithms and “additive secret sharing” provide an exact solution for the non-federated problem? This should be discussed in the manuscript.

The manuscript does not adequately describe “additive secret sharing.” It could be explicitly illustrated in Fig. 1, and a higher-level explanation should be included in the methods section of the main text.

The sentence, “Due to privacy regulations, finding publicly available multicenter patient-derived data suitable for evaluating FedProt was challenging,” is somewhat ambiguous. It gives the impression that FedProt might not comply with privacy regulations. Perhaps the authors mean that benchmarking cannot be performed on privacy-sensitive data due to the need for data pooling in the non-federated DEqMS. This should be clarified.

The manuscript does not explore other proteomics analysis task could be performed under privacy constraints. For example, could federated PCA be used to detect outliers or batch effects? Could these or similar steps be incorporated into FedProt?

Minors

Lines 64-80 are quite technical and full of abbreviations. Authors could simplify the text for a more general/non-proteomics readership; and stick to most important abbreviations, i.e. DDA vs DIA.

Line 194 - “We assumed that collaborating parties could agree on using a uniform data preprocessing protocol.” Shouldn't a uniform pre-processing protocol be crucial for such analysis? Maybe a more explicit sentence here would be a better fit? “FedProt requires that all parties agree ... “

Line 459-460 - these statements are unclear. Please consider rewriting.

Reviewer #3 (Remarks to the Author):

The authors of this study developed FedProt — a tool utilizing federated learning for multi-centered data integration. Proteomics data generated in this study is from five different institutions using different mass spectrometers. Moreover, the pooled data combined with DEqMS analysis results were used as the ground truth for the comparison between different methods. In various comparative analyses, FedProt consistently outperformed other meta-analysis methods in terms of accuracy. The development of FedProt indeed addresses a significant issue in proteomics data analysis. However, the analytical results and comparison methods presented by the authors have certain limitations. Below are some suggestions:

1. The authors mentioned in lines 194-195, “We assumed that collaborating parties could agree on using a uniform data preprocessing protocol”. However, when collecting data from multiple centers, each center may use different protocols. To increase the usability, it would be beneficial to assess the performance of FedProt with data processed using different preprocessing protocols.

2. Since FedProt can integrate data from multiple centers, could it also be used to integrate data from different studies? This would be particularly practical for research groups wishing to use data from databases such as PRIDE. It could be extremely beneficial for rare disease studies that may struggle to collect enough samples from a single center. However, as mentioned in point 1, these studies may not use identical preprocessing protocols. I would suggest that the authors present some results demonstrating the effectiveness of FedProt in handling such cases.

3. Different centers may use different quantification software, such as MaxQuant and Proteome Discoverer for DDA, or MaxDIA and DIA-NN for DIA. These tools can produce different results in protein quantification and identification. Additionally, many centers may only store the quantification tables for long-term storage, rather than the raw data. It would be helpful to demonstrate the performance of FedProt in integrating quantification tables from different software.

4. Since statistical significance is based on the adjusted p-value, it would be interesting to investigate the results using different cutoffs for log₂ fold-change. The authors set different cutoffs for various analyses and datasets, such as human serum, bacterial data, and data imbalance comparisons. Is it possible that different meta-analysis methods may show varying accuracy based on different log₂ fold-change cutoffs?

5. It would be very helpful for readers to understand the workflow, including normalization, filtering, and data integration, by

using a simple example or figure like Figure S11.

6. To help readers understand the differences between FedProt and other meta-analysis methods, it would be helpful to generate a figure comparing the methods, similar to Flimma (Fig 1, <https://genomebiology.biomedcentral.com/articles/10.1186/s13059-021-02553-2>).

7. Is DIA-NN run using a "library-free search" or a "DDA spectra library-dependent" approach? Please address this in the manuscript, as it may influence the accuracy of protein identification and quantification.

8. I suggest the authors carefully reconsider the naming conventions for the tables and figures. For example, the column names in Table S2 are quite confusing. From my understanding, B1, B2, and B3 should refer to Batch 1, 2, and 3, not condition B.

9. In Figure 2a and c, the y-axis is labeled "other methods," which feels a bit odd. Would it be better to change it to "FedProt and other meta-analysis methods"?

10. In Figure 2b, the labels on the x-axis are not capitalized. It would be better to make them consistent with Figure 2D.

11. Line 246. "Tables 2" should be "Table 2".

Reviewer #3 (Remarks on code availability):

1. The code is well-written, but the documentation regarding Docker is unclear. It may need to run Docker when executing FedProt, but this is not mentioned. Users who are not familiar with Docker may encounter difficulties in executing or installing the program.

2. Having a log file to store the executed commands and the process will be a great help for users to review or troubleshoot issues.

3. In "Prerequisite" section of README, There is a typo in the comment of the second command - # first, create and go "the" the dir, where test folder will be created. It should be - # first, create and go "to" the dir, where test folder will be created.

Version 1:

Decision Letter:

Our ref: NATCOMPUTSCI-24-2014A

14th April 2025

Dear Dr. Burankova,

Thank you for submitting your revised manuscript "Privacy-Preserving Multi-center Differential Protein Abundance Analysis with FedProt" (NATCOMPUTSCI-24-2014A). It has now been seen by the original referees and their comments are below. The reviewers find that the paper has improved in revision, and therefore we'll be happy in principle to publish it in Nature Computational Science, pending minor revisions to satisfy the referees' final requests and to comply with our editorial and formatting guidelines.

TRANSPARENT PEER REVIEW

Nature Computational Science offers a transparent peer review option for original research manuscripts. We encourage increased transparency in peer review by publishing the reviewer comments, author rebuttal letters and editorial decision letters if the authors agree. Such peer review material is made available as a supplementary peer review file. **Please remember to choose, using the manuscript system, whether or not you want to participate in transparent peer review.**

Please note: we allow redactions to authors' rebuttal and reviewer comments in the interest of confidentiality. If you are concerned about the release of confidential data, please let us know specifically what information you would like to have removed. Please note that we cannot incorporate redactions for any other reasons. Reviewer names will be published in the peer review files if the reviewer signed the comments to authors, or if reviewers explicitly agree to release their name. For more information, please refer to our <https://www.nature.com/documents/nr-transparent-peer-review.pdf> target="new">FAQ page.

Thank you again for your interest in Nature Computational Science. Please do not hesitate to contact me if you have any questions.

Sincerely,

Ananya Rastogi, PhD
Senior Editor
Nature Computational Science

ORCID

Reviewer #1 (Remarks to the Author):

The authors have answered all my concerns.

Reviewer #1 (Remarks on code availability):

I have navigated and inspected the code but did not try to run it myself.

Reviewer #2 (Remarks to the Author):

The authors did a very good job in answering all my requests.

Reviewer #3 (Remarks to the Author):

The author has thoroughly answered all my questions and provided very impressive figures. Additionally, they have included some FedProt workflows, comparisons, and explanatory charts, which enhance readability. I am satisfied with this and have no further questions.

Reviewer #3 (Remarks on code availability):

The author has fixed the issues I raised in the program and also has generated log folder for capturing all processing steps.

Version 2:

Decision Letter:

Dear Ms Burankova,

We are pleased to inform you that your Resource "Privacy-Preserving Multi-center Differential Protein Abundance Analysis with FedProt" has now been accepted for publication in Nature Computational Science.

Once your manuscript is typeset, you will receive an email with a link to choose the appropriate publishing options for your paper and our Author Services team will be in touch regarding any additional information that may be required.

Acceptance of your manuscript is conditional on all authors' agreement with our publication policies (see <https://www.nature.com/natcomputsci/for-authors>). In particular your manuscript must not be published elsewhere and there must be no announcement of the work to any media outlet until the publication date (the day on which it is uploaded onto our web site).

Before your manuscript is typeset, we will edit the text to ensure it is intelligible to our wide readership and conforms to house style. We look particularly carefully at the titles of all papers to ensure that they are relatively brief and understandable.

Once your manuscript is typeset, you will receive a link to your electronic proof via email with a request to make any corrections within 48 hours. If, when you receive your proof, you cannot meet this deadline, please inform us at rjsproduction@springernature.com immediately.

If you have queries at any point during the production process then please contact the production team at rjsproduction@springernature.com.

We welcome the submission of potential cover material (including a short caption of around 40 words) related to your manuscript; suggestions should be sent to Nature Computational Science as electronic files (the image should be 300 dpi at 210 x 297 mm in either TIFF or JPEG format). We also welcome suggestions for the Hero Image, which appears at the top of our [home page](http://www.nature.com/natcomputsci); these should be 72 dpi at 1400 x 400 pixels in JPEG format. Please note that such pictures should be selected more for their aesthetic appeal than for their scientific content, and that colour images work better than black and white or grayscale images. Please do not try to design a cover with the Nature Computational Science logo etc., and please do not submit composites of images related to your work. I am sure you will understand that we cannot make any promise as to whether any of your suggestions might be selected for the cover of the journal.

Best regards,

Ananya Rastogi, PhD
Senior Editor
Nature Computational Science

P.S. Click on the following link if you would like to recommend Nature Computational Science to your librarian: <https://www.springernature.com/gp/librarians/recommend-to-your-library>

** Visit the Springer Nature Editorial and Publishing website at <http://editorial-jobs.springernature.com> for more information about our career opportunities. If you have any questions please click [here](mailto:editorial.publishing.jobs@springernature.com).**

Response letter to reviewers

We would like to thank the reviewers and the editor for the thoughtful comments and suggestions provided. These comments were very helpful for improving the manuscript.

Below, each concern raised is presented verbatim *in blue italics*, followed by our detailed responses in black. All citations from the main text are *italicized*, and textual revisions made to the manuscript are highlighted *in red italics* here (and marked in green in the manuscript).

The points raised by the editor

Each concern raised by the reviewers is assigned a unique reference (e.g., R1.1, R1.2, etc.). When the editor's comment aligns with the reviewer's comment, the same reference number applies. Detailed responses addressing each of these concerns can be found in the corresponding response sections later in this document, under the same reference numbers.

(R1.1) - Most systems used for the handling of sensitive data do not have access to the internet, or the connection is very restricted. Please address how this is handled.

(R1.3) - Please discuss how the performance scales with the number of samples.

(R2.2) - Please describe "additive secret sharing."

(R2.3) - The sentence, "Due to privacy regulations, finding publicly available multicenter patient-derived data suitable for evaluating FedProt was challenging," is somewhat ambiguous. It gives the impression that FedProt might not comply with privacy regulations. This should be clarified.

(R2.4) - Please explore other proteomics analysis task that could be performed under privacy constraints.

(R3.1) - To increase the usability, it would be beneficial to assess the performance of FedProt with data processed using different preprocessing protocols.

(R3.3) - Please demonstrate the performance of FedProt in integrating quantification tables from different software.

(R3.4) - Since statistical significance is based on the adjusted p-value, it would be interesting to investigate the results using different cutoffs for log2 fold-change.

Additionally, we have made a correction to the bacterial data analysis. Four technical QC samples are now removed from differential analysis of the bacterial dataset. This correction resulted in minor changes of the results (highlighted in the main text) that did not change the conclusions.

We have updated the PRONE R package reference since it was published as a preprint. The author list has been updated to reflect the correct names: Christine von Toerne (previously misprinted as Christine von Törne), Teresa K. Barth and Stefan F. Lichtenthaler (now including the middle initial).

Reviewers comments:

Reviewer #1 (Remarks to the Author):

In this manuscript, Burankova et al. present a solution for federated quantitative proteomic analysis. The problem of jointly analyzing proteomic data is increasing as proteomics becomes available for cohorts. Overall, the work is well executed and the paper well written. I have mainly suggestions regarding points of discussion but otherwise recommend the publication of this paper.

Main comments

R1.1 - Most systems used for the handling of sensitive data do not have access to the internet, or the connection is very restricted. How is this handled by the author's approach? What are the requirements of the proposed approach in terms of software environment? How would their system work on a setup like the one of the UK biobank?

We would like to thank the reviewer for this important question. Raw MS data are preprocessed (quantified) locally, so there is no need to transfer large raw files over the internet. FedProt requires only the resulting PG × samples (and PG × counts) matrices, which must be located on a machine with internet access. Python and Docker must be installed.

Our setup requires a standard computing environment (Windows, Mac, Linux) that supports Docker containers, as FeatureCloud operates within Docker. To run the analysis in its current configuration, internet access, registration via email, and logging in on the platform are required.

We should also note that while our current implementation relies on an internet connection to coordinate on the FeatureCloud platform, the system supports deployment on secure, pre-approved channels or even on a completely private network (Matschinske et al. 2023). For instance, in environments like the UK Biobank, where internet access might be restricted, one can adjust the configuration to use a private relay server instance by modifying the controller's configuration file. This setup ensures that all communications remain within the secured network and traffic is shielded from anyone outside the collaboration.

We have revised the manuscript text:

“To make this decentralized approach and its complex infrastructure available to a broad community, we implemented FedProt as a web-based app with a user-friendly graphical interface. FedProt is published as a certified app in the FeatureCloud³⁴ app store with documentation and quick-start guidelines (<https://featurecloud.ai/app/fedprot>). While this implementation relies on an internet connection for FeatureCloud coordination, it is fully adaptable. FedProt can be reconfigured to operate on secure, private networks or pre-approved communication channels³⁴, ensuring that all data communications remain within controlled institutional environments.”

In principle, modifying the FedProt method to run on machines even without internet access is possible. In such scenarios, only intermediate data would need to be transferred, for example, via a portable data drive (USB stick, CD, etc), to a machine that can communicate with the coordinator. However, this way, it is impossible to fully automate client communication with the coordinator, as is possible within the FeatureCloud platform.

As FedProt is open source and licensed under Apache 2.0, users are free to modify and adapt the code even for commercial usage, offering additional flexibility for adapting to different computing environments.

FedProt was tested on Ubuntu, Mac, and Windows. It requires access to the Internet, Python, and Docker installed to run FeatureCloud client applications. For optimal performance, using machines with at least 4 GB of RAM is recommended. We have updated the Prerequisite section on the FedProt GitHub page (<https://github.com/Freddsle/FedProt?tab=readme-ov-file#prerequisite>). Detailed installation instructions and configuration guidelines are available on the FedProt page (<https://featurecloud.ai/app/fedprot>) and in the FeatureCloud documentation (<https://featurecloud.ai/developers> and <https://featurecloud.ai/researchers>). We hope this clarifies our approach and adequately addresses the questions raised.

R1.2 - As the authors noticed, getting access to data is hard. In fact, experience says that in meta-analyses coordination of analysts is more of a problem than data processing. And if the protocol is too much of a burden for the analyst, the meta-analysis will simply never happen. Can the authors comment on the ease of their method in comparison to a standard meta-analysis?

Our approach addresses not only the problem of data access but also simplifies both the coordination among analysts and the data processing workflow compared to traditional meta-analyses. Traditional meta-analyses are often affected by difficulties such as incomplete retrieval of individual participant data (IPD) and selective reporting, issues that introduce systematic biases and complicate the analysis (Speich et al. 2022; Polanin 2018).

In contrast, our method overcomes these problems by using federated learning and eliminates the need for centralized aggregation and storage of raw data. FedProt requires only standard outputs, specifically, PGs x samples and PGs x counts matrices, generated by widely used software (e.g., MaxQuant, Spectronaut, DIA-NN) agreed upon between the participants. Thus, for FedProt, an analyst only needs to preprocess (quantify and FDR-filter) raw MS data in a way that is common in proteomics. This minimizes the burden of extensive data preparation while ensuring consistency through the use of consistent software and libraries (FASTA or pre-generated spectral libraries).

The implementation of FedProt as an app on the FeatureCloud platform further enhances the workflow by automating the analysis process. After the coordinator sends out invitations, participants can simply join and start the analysis. The process then proceeds automatically, eliminating the sequential data processing delays and communication challenges typical of conventional meta-analyses. Each participant receives a final result table, allowing independent follow-up without synchronization issues.

In addition, because no raw data is shared, our federated approach eases ethical, legal, and proprietary concerns, thus facilitating collaborations not only among academic institutions but also with industry partners, including pharmaceutical companies with strict data privacy requirements.

These points are discussed in the subsequent sections of the manuscript:

“Meta-analysis performance improves with an increase in sample sizes and number of studies¹⁷ or with the availability of raw data for combined re-analysis¹⁸, which is challenging in proteomics. ... Additionally, meta-analyses face challenges related to heterogeneity from variations in experimental design, sample characteristics, and equipment for peptide separation and MS data acquisition²⁵. They cannot fully account for cohort differences, such as target class imbalance or covariate distribution variations^{26,27}. Differences in data processing steps, such as normalization, may also significantly impact the results²⁸.”

*“We assumed that collaborating parties could agree on using a uniform data preprocessing protocol. Although FedProt can tolerate minor variations in preprocessing if a strictly uniform protocol is not feasible, our evaluations on balanced dataset (see **Supplementary Results** section) indicate that results from non-uniform preprocessing are similar to those obtained by centralized DEqMS analysis. Nevertheless, we strongly recommend that all parties use the same quantification software for preprocessing whenever possible.”*

“FedProt demonstrated resilience to a sample size imbalance between cohorts, the ability to work with data with missing values, and accounting for batch effects.”

“Before the federated analysis, each participant preprocesses its local dataset independently FedProt requires ...: a protein intensities matrix. ... Currently, two types of normalization are implemented.”

R1.3 - The sample sizes proposed by the authors for testing are very small, modern cohort proteomic data feature several hundreds or thousands of samples. How does the performance scale with the number of samples? Will the gain of running a federated analysis over a standard meta-analysis still be significant?

We agree that modern MS-based proteomics cohorts design are shifting towards larger sample sizes, typically ranging from several hundred to a few thousand samples. However, large-scale multi-center MS-based datasets still remain relatively uncommon compared to antibody-based platforms such as Olink (Eldjarn et al. 2023). In fact, the largest MS-based datasets currently available include only hundreds to low thousands of samples, with the field still progressing toward truly large-scale MS-based proteomics. Moreover, MS-based proteomics studies, particularly in a multi-center context or during ring trials, are often limited to a few hundred samples per cohort. As the reviewer noted, the actual cohorts of the real datasets we used are not as large as we would like. FedProt is designed to help researchers to increase cohort size while preserving data privacy to help overcome this limitation in MS-based proteomics studies. One of the main steps of the FedProt method is federated linear regression (Karr et al. 2005), which is designed to replicate the results of a centralized analysis exactly, regardless of the number of samples. To illustrate this, we performed simulation experiments with 66, 660, and 6600 samples (**Supplementary Table S8**). In each case, we keep a constant set of 6000 proteins with approximately 350 differentially expressed proteins (subject to random variation). Our results (see **Supplementary Fig. S18** and **Supplementary Table S9**) demonstrate that FedProt’s results remain consistent with those from centralized analysis, even as cohort size increases.

Table S8. Characteristics of the simulated datasets used to evaluate the performance scaling with the number of samples. Number of samples in each cohort (C1-C3) in each condition (A, B), for confounder column — proportion of samples among condition B samples.

Dataset	Total samples	Cohorts			Condition A			Condition B			in B — frequency of samples with the confounder		
		C1	C2	C3	C1	C2	C3	C1	C2	C3	C1	C2	C3
Balanced													
1x	66	22	22	22	11	11	11	11	11	11	0.6	0.6	0.6
10x	660	220	220	220	110	110	110	110	110	110			

100x	6600	2200	2200	2200	1100	1100	1100	1100	1100	1100			
Strongly imbalanced													
1x	66	12	8	46	9	3	31	3	5	15	0.2	0.5	0.7
10x	660	45	94	521	37	31	304	8	63	217			
100x	6600	450	940	5210	370	310	3040	80	630	2170			

Figure S18. Ranking consistency between centralized and decentralized methods for simulated datasets under different cohort sizes.

Panel a – simulated balanced datasets (1x, 10x, and 100x), panel b – simulated imbalance datasets (1x, 10x, and 100x). The dependency of the Jaccard similarity coefficient on the number of top-ranked proteins identified by the centralized *DEqMS* and decentralized approaches. Proteins were ranked based on their decreasing negative log-transformed BH-adjusted p-values and not filtered by log₂FC. The generation of simulated data and the subsequent data analysis were repeated 10 times and aggregated results reported.

Table S9. Mean and maximum absolute differences in negative log-transformed adjusted p-values, Jaccard similarity coefficients, and error rates (false positives and false negatives) for the results of FedProt and selected meta-analysis approaches compared to centralized *DEqMS* results using simulated datasets. Jaccard similarity coefficients and error rates were computed with $|\log_2\text{FC}| > 1$ and adj. p-value < 0.05 thresholds. The values corresponding to the best performance between all methods are highlighted in bold font. The generation of simulated data and the subsequent data analysis were repeated 10 times — mean and standard deviation are reported.

Dataset	Method	Mean difference	Maximal difference	FP	FN	Jaccard similarity coefficient
Balanced						

Balanced 1x	FedProt	8.93E-16 ± 1.19E-16	1.44E-14 ± 5.58E-15	0.0 ± 0.0	0.0 ± 0.0	1.00 ± 0.0
	Fisher	0.05 ± 0.00	1.52 ± 0.38	0.20 ± 0.42	6.90 ± 4.48	0.17 ± 0.21
	Stouffer	0.06 ± 0.01	1.57 ± 0.33	0.0 ± 0.0	7.10 ± 4.46	0.16 ± 0.18
	REM	0.04 ± 0.00	1.82 ± 0.54	3.30 ± 1.64	1.80 ± 1.48	0.55 ± 0.17
	RankProd	0.67 ± 0.01	6.68 ± 1.29	237.00 ± 17.20	0.0 ± 0.0	0.04 ± 0.02
Balanced 10x	FedProt	3.49E-15 ± 2.31E-16	2.08E-13 ± 7.19E-14	0.0 ± 0.0	0.0 ± 0.0	1.00 ± 0.0
	Fisher	0.13 ± 0.01	13.80 ± 4.17	1.30 ± 0.95	1.50 ± 1.08	0.94 ± 0.03
	Stouffer	0.14 ± 0.01	10.30 ± 3.02	1.30 ± 0.95	1.70 ± 1.16	0.93 ± 0.03
	REM	0.16 ± 0.01	20.50 ± 3.37	3.70 ± 1.95	5.00 ± 1.89	0.81 ± 0.05
	RankProd	0.81 ± 0.02	33.20 ± 5.88	5.00 ± 2.21	0.50 ± 0.53	0.88 ± 0.04
Balanced 100x	FedProt	1.77E-14 ± 8.88E-16	1.36E-12 ± 2.75E-13	0.0 ± 0.0	0.0 ± 0.0	1.00 ± 0.0
	Fisher	0.55 ± 0.06	153.00 ± 38.30	1.80 ± 1.69	0.50 ± 0.85	0.92 ± 0.07
	Stouffer	0.47 ± 0.04	116.00 ± 27.90	1.80 ± 1.69	0.50 ± 0.85	0.92 ± 0.07
	REM	1.39 ± 0.13	206.00 ± 26.10	3.20 ± 2.15	1.90 ± 1.37	0.82 ± 0.06
	RankProd	4.92 ± 0.19	294.00 ± 4.44	1.90 ± 1.79	0.50 ± 0.85	0.92 ± 0.07
Imbalanced						
Imbalanced 1x	FedProt	1.61E-15 ± 2.08E-16	2.26E-14 ± 5.88E-15	0.0 ± 0.0	0.0 ± 0.0	1.00 ± 0.0
	Fisher	0.05 ± 0.00	1.56 ± 0.32	0.0 ± 0.0	9.30 ± 4.55	0.04 ± 0.05
	Stouffer	0.06 ± 0.00	2.04 ± 0.52	0.0 ± 0.0	9.70 ± 4.81	0.01 ± 0.02
	REM	0.05 ± 0.00	2.38 ± 0.77	2.60 ± 1.58	4.80 ± 4.64	0.41 ± 0.18
	RankProd	0.66 ± 0.01	6.86 ± 1.21	269.00 ± 9.96	0.70 ± 0.68	0.03 ± 0.02
Imbalanced 10x	FedProt	1.48E-14 ± 1.78E-15	5.46E-13 ± 1.07E-13	0.0 ± 0.0	0.0 ± 0.0	1.00 ± 0.0
	Fisher	0.15 ± 0.01	14.40 ± 4.70	20.50 ± 4.28	16.20 ± 4.24	0.43 ± 0.04
	Stouffer	0.32 ± 0.01	17.90 ± 2.47	18.60 ± 4.06	17.00 ± 4.67	0.43 ± 0.05
	REM	0.24 ± 0.01	30.50 ± 4.47	3.20 ± 1.23	7.50 ± 3.24	0.78 ± 0.06
	RankProd	0.95 ± 0.03	38.20 ± 3.93	71.10 ± 13.50	15.90 ± 4.41	0.25 ± 0.02
Imbalanced 100x	FedProt	7.90E-14 ± 5.69E-15	5.99E-12 ± 7.93E-13	0.0 ± 0.0	0.0 ± 0.0	1.00 ± 0.0
	Fisher	0.69 ± 0.08	231.00 ± 42.10	16.50 ± 3.63	9.40 ± 3.60	0.40 ± 0.08
	Stouffer	1.86 ± 0.06	112.00 ± 14.40	16.50 ± 3.63	9.40 ± 3.60	0.40 ± 0.08
	REM	2.62 ± 0.09	252.00 ± 17.90	5.10 ± 2.28	4.30 ± 1.83	0.70 ± 0.06
	RankProd	5.80 ± 0.15	296.00 ± 0.17	17.20 ± 3.29	9.40 ± 3.60	0.40 ± 0.08

Indeed, when the sample size increases, meta-analyses show improved performance (**Supplementary Fig. S18** and **Supplementary Table S9**). For all meta-analysis methods tested, the small cohort size resulted in different results from the central analysis regardless of whether the dataset was balanced or imbalanced. The large sample size (x100, 6600 samples) allowed meta-analyses to accurately sort the top 350 proteins (**Supplementary Fig. S18**, the number of proteins in which differential expression was introduced). However, when considering all proteins and applying logFC and p-value simultaneously (**Supplementary Table S9**), the results of meta-analyses still differed from the central analysis, and this discrepancy is particularly pronounced in imbalanced datasets.

In conclusion, our results demonstrate that FedProt scales effectively and maintains stable performance across a range of cohort sizes. Importantly, while meta-analyses benefit from larger sample sizes, they do not fully replicate centralized results in more complex data scenarios, particularly in the presence of imbalance. This indicates that the gain from running a federated analysis over a standard meta-analysis remains significant even if sample size is big,

not only in achieving consistency with centralized results but also in ensuring robust performance under diverse conditions.

We added the analysis described here as a Supplementary Results section. Additionally, we modified the Result section of the manuscript by adding:

“Additionally, we evaluated FedProt's scalability using simulated datasets with 66, 660, and 6600 samples (see Supplementary Results). The results confirmed that FedProt consistently replicates centralized analysis results across varying cohort sizes, whereas meta-analyses, even with larger sample sizes, do not fully reproduce the central results, especially under data imbalance. These findings further highlight the robustness and scalability of FedProt over standard meta-analysis approaches.”

Minor writing details

R1.4 - *In the introduction, avoid considerations on ‘comprehensiveness’. Also avoid ranking methods or having judgment on which is best - this is not the right place for such considerations.*

We appreciate the reviewer’s feedback and have revised the language in the Introduction. To avoid considerations of “comprehensiveness” and ranking methods, we have replaced subjective qualifiers with neutral descriptions:

- 1) *“(MS)-based proteomics offers ~~comprehensive~~**detailed** insight into dynamic protein composition, interactions, and modifications not **readily** inferred solely from genomics or transcriptomics data”,*
- 2) *“~~DIA systematically fragments all ions within predefined m/z windows, ensuring comprehensive and unbiased peptide quantification and identification.~~ **By systematically fragmenting all ions within predefined mass ranges, DIA ensures broad and unbiased peptide identification**”,*
- 3) *“~~TMT labeling~~**This approach** allows simultaneous comparison of peptide abundances across multiple samples in a single MS run. ~~and provides the highest accuracy of all relative quantitative proteomic techniques~~”*

R1.5 - *When considering data privacy, please avoid mixing legal considerations with potential threats. Rare variant identification, reidentification, reconstruction are not the reason why privacy should be preserved. The data are private, by law, they should be handled as such. If it would be a standalone birth year, albeit totally harmless these data would still need to be handled in a privacy-preserving system.*

We thank the reviewer for the comment. In response, we have revised the text to separate the legal obligations from technical considerations. The revised text:

*“To maximize clinical proteomics' potential, analyzing larger multi-center patient cohorts is necessary to increase statistical power and achieve more robust results, especially for identifying rare disease subtypes^{11,12}. However, integrating patient-derived MS data **and proteomics profiles** distributed across multiple research institutions can be problematic due to privacy concerns, **as they are legally classified as private and must be handled accordingly**¹³. Similar to transcriptomics, proteomics data can uncover rare sequence variants¹⁴ or be subject to genotype reconstruction attacks¹⁵. ~~Therefore, raw patient-derived MS data and proteomics profiles must be treated as confidential.~~”*

R1.6 - Please provide basic summary statistics on the result of the proteomic quantitative analyses.

For the bacterial and human serum datasets, the number of proteins identified across all samples is summarized in Supplementary Figure S3. Descriptive statistics, including density plots and boxplots for each center, are provided in Supplementary Figure S1. Additionally, the results of the differential abundance analysis, including $|\log_2FC|$ values, p-values, and adjusted p-values computed using all the methods applied in this study (DEqMS, FedProt, and meta-analysis approaches), are detailed in the tables available on GitHub <https://github.com/Freddsle/FedProt> (inside the evaluation folder at 'evaluation/[dataset name]/[subfolder]/results').

Figure S1. Descriptive statistics and principal component analysis (PCA) plots for the bacterial (panels a-d) and human serum (panels e-h) datasets.

Intensity density plots (panels a, e) are shown for log₂-transformed data, colored by the lab for the bacterial dataset (panels a and b) and by centers and TMT pools for the human serum dataset (panels e and f). PCA plots before (as used in FedProt, panels c and g) and after batch effect correction (panels d and h) using limma removeBatchEffect function for bacterial dataset including QC samples (panels c and d) and for the human serum dataset (panels g and h). QC samples in the bacterial dataset represent technical replicates generated for QC purposes and were removed from the dataset during FedProt evaluation.

Figure S3.

The number of protein groups that could be analyzed by the DEqMS method inside each lab separately. Panel a — for the bacterial dataset; Panel b — for the human serum dataset.

R1.7 - In Figure 1, please clearly highlight that the personal-level information is contained in the local servers and are never shared. This could also be made clearer in the text.

Thank you for this suggestion. We have revised Figure 1 and its description, highlighting that the personal-level information and client's local parameters are never shared. Main changes:

- 1) We added panel B with a visual explanation of additive secret sharing.
- 2) To the description of data preparation step on panel A, we have added "*Personal-level information (proteomics profiles) is contained in the client's servers and are never shared (dashed lines around clients highlight different physical locations).*"
- 3) On panel B, we added that "*no single party can reconstruct the unmasked data*".

Figure 1. FedProt Workflow Overview.

Panel a – Federated workflow overview:

1) Data Preparation: Data owners collect and preprocess MS data, obtain protein intensity and peptide count matrices, and define design matrices before participating as clients. **Personal-level information (proteomics profiles) is contained in the client's servers and are never shared (dashed lines around clients highlight different physical locations).**

2) Federated Learning: Clients communicate with the central server (coordinator) to collaboratively train a global model without revealing their individual datasets, but through the exchange of local model parameters. The clients protect their local parameters using additive secret sharing (blue arrows), **where encrypted parts of masked data are exchanged among clients (blue arrows), ensuring that no single party receives more than one piece of the data from each of the other clients.** In case the data are not numeric, such as protein group names, they are sent to the coordinator without additive secret sharing (green arrows). The coordinator returns updated global parameters to clients (black arrows).

3) Result: After all federated computations, all clients receive the results mathematically equivalent to the results of centralized analysis of pooled dataset with *DEqMS* formatted as a table with abundance fold-changes, confidence intervals, and adjusted p-values.

Panel b: Overview of data communication using SMPC (additive secret sharing) inside FedProt:

The clients protect their local parameters using additive secret sharing. Each client data is masked with a noise mask. The noisy data and the noise masks are splitted into n encrypted parts ($n > 2$). These parts are exchanged among clients (blue arrows) via a relay server, ensuring that no single party receives more than one piece of the data from each of the other clients. After decrypting the received parts, clients sum the data, and send the re-encrypted sums to the coordinator, who decrypts and aggregates the sums to compute the global result. For details see Methods and Supplementary methods.

R1.8 - Please define what 'masks' refer to in the context of the paper.

Thank you for this suggestion. We have revised Figure 1 by adding more details about additive secret sharing, as well as an introduction where noise masks in additive secret sharing are introduced. Thus, when we talk about noise masks in the context of additive secret sharing, we use “noise masks”, and when (in Methods) we describe a mask for federated linear regression, we use “design mask D”.

*“To further enhance privacy and protect local data from reconstruction attacks, we use additive secret sharing²⁹. In this method, each client generates multiple **noise** masks and communicates these masks and masked data to the other parties, ensuring no single party can reconstruct the unmasked data (blue arrows **and the right panel**, Fig. 1). ... This scheme allows global aggregation of local results without revealing any local values, thereby enhancing privacy compared to a pure federated learning scheme (see Methods for further details).”*

Reviewer #1 (Remarks on code availability):

I have inspected the code but did not try to install or run the application.

The repository is very well documented. The code is clear and structured.

Reviewer #2 (Remarks to the Author):

In this manuscript, Burankova and colleagues describe a federated learning approach for protein abundance analysis. Federated learning is a critical methodology for working with sensitive patient data, as it enables data analysis without the need to pool sensitive information at a single site. For this, the authors propose the use of federated learning with multi-party computation. This approach mirrors a method previously proposed by the same authors for analyzing gene expression data. A novel aspect of this work is its consideration of proteomics-specific challenges, such as the measurement of distinct proteins and platform-specific artifacts.

Overall, this is a relevant and sound approach. The authors conducted evaluation experiments of data from two distinct platforms (DDA and DIA); and simulated data for robustness analysis regarding batch artefacts and class imbalance. Results indicate that FedProt achieves comparable or identical performance to the non-federated version and that it outperforms competing methods, which are primarily based on meta-analysis. The manuscript is well written, but a few recommendations could further improve its quality.

Specific points

***R2.1** - Results for FedProt (Fig. 2 and Table 3) appear very similar to the non-federated DEqMS. Could this be due to the fact that federated learning algorithms and “additive secret sharing” provide an exact solution for the non-federated problem? This should be discussed in the manuscript.*

Indeed, FedProt yields results nearly identical to the non-federated, centralized DEqMS analysis because our federated approach employs precisely the same linear modeling and variance estimation steps as DEqMS, but executes them in a distributed, privacy-preserving manner using additive secret sharing.

More specifically, FedProt uses a federated linear modeling approach (Karr et al. 2005) that maintains the same intermediate statistics as if the data were centralized. The subsequent steps, including the empirical Bayes step, p-values calculation, and DEqMS's peptide-count adjustment, are carried out by the coordinator as they would be in a single-site DEqMS analysis.

Additive secret sharing ensures each site's raw data are locally split into “shares” and encrypted, then securely aggregated by the relay server. Once aggregated, the coordinator obtains exactly the same global sums used by DEqMS, without ever accessing or exposing any site's raw data.

Hence, the final parameter estimates (fold changes, p-values) are mathematically the same as those of the non-federated DEqMS scenario, as illustrated in Figure 2 and Tables 2 and 3 of the manuscript. The negligible differences arise from the floating-point arithmetic differences between R and Python. Thus, the near-identical performance of FedProt and DEqMS results highlights that FedProt reproduces the standard centralized computation, rather than approximating it, all while preserving patient privacy.

We have clarified it in the revised manuscript (new / modified parts are marked in red):

1) Results: “*FedProt represents the mathematical equivalent of DEqMS, the accurate variance estimation workflow for mass spectrometry-based proteomics data. ... The FedProt approach allows us to obtain the same result as centralized pooled data analysis while implementing strong privacy-preserving measures, ensuring no patient-level data is shared and exchanged parameters are **hidden from other participantsmasked**.*”

2) Discussion:

“Our evaluation confirmed that FedProt's results are equivalent to those from the original DEqMS method when applied to centralized and pooled data. In all tests, particularly in identifying top differentially abundant proteins, FedProt's results consistently matched the centralized approach results, surpassing all tested meta-analysis methods. Additionally, FedProt demonstrated resilience to a sample size imbalance between cohorts, the ability to work with data with missing values, and accounting for batch effects.

The results from FedProt and centralized DEqMS analysis are nearly identical because the federated approach employs the same linear modeling and variance estimation steps as centralized DEqMS, with additive secret sharing enabling the secure aggregation of local intermediate statistics. The minor differences observed are attributable solely to floating-point arithmetic variations between R (limma/DEqMS) and Python (FedProt). “

R2.2 - *The manuscript does not adequately describe “additive secret sharing.” It could be explicitly illustrated in Fig. 1, and a higher-level explanation should be included in the methods section of the main text.*

Additive secret sharing was explained in supplementary text and illustrated in Supplementary Figure S10 (Fig. S19 in the revised version). As recommended, we added more details to the description of additive secret sharing in Methods of the main text:

“To minimize the risk of reconstruction attack, $(X^i)^T X^i$, $(X^i)^T Y^i$, as well as any local computation result shared with the aggregating server, are protected by additive secret sharing²⁹. Briefly, each client generates n randomly sampled noise masks, r_1, \dots, r_n , as equally distributed random values, and calculates corresponding noisy data as $(M - r_1 - \dots - r_n)$. This noisy data, alongside the noise masks (divided into n pieces in total), is communicated with other computational parties via a relay server, ensuring no party receives more than one piece (one data piece and one noise piece) from any specific client. The data pieces are encrypted with each party's public key to ensure the data cannot be intercepted. Once the encrypted pieces are received, each party decrypts and sums the received data, then sends the re-encrypted received sums to the coordinator again via the relay server. The coordinator gets the summed parts and, in case of sending $(X^i)^T X^i$ and $(X^i)^T Y^i$, obtains the global coefficients $\hat{\beta}$. ~~Each client generates n randomly sampled masks, r_1, \dots, r_n , as equally distributed random values, and calculates the masked data $(M - r_1 - \dots - r_n)$. This noisy data, alongside the masks (n pieces in total), is communicated with other computational parties while ensuring no party receives more than one piece of data from any specific client. The data pieces are encrypted with each party's public key to ensure the data cannot be intercepted. Each party sums the received pieces and sends the results to the coordinator. The coordinator gets the summed parts and obtains the global coefficients $\hat{\beta}$.~~

~~During the additive secure aggregation process, the noises cancel out due to their additive nature, resulting in the correct global aggregation of local parameters without any noticeable impact on the final outcome compared to non-secure aggregation. This cancellation mechanism preserves data privacy, as the individual intermediate results remain are not revealed to both the coordinator and any other parties. During the additive secure aggregation process the noises will be canceled without noticeable impact on the outcome in comparison with non-secure aggregation while it does not reveal the clients' intermediate results to the coordinator or any other parties. By increasing the number of parties, the risk of collusion to~~

reconstruct the clients' intermediate results will be further reduced. To simplify the technical aspects of communicating data, FeatureCloud³³ passes all data through the relay server which cannot decrypt the data, see details in Supplementary methods."

We have also revised Fig. 1 and believe that these changes address the reviewer's concerns and improve the clarity and completeness of our manuscript. Main changes of Fig. 1:

- 1) We added panel B with a visual explanation of additive secret sharing.
- 2) To the description of data preparation step on panel A, we have added "*Personal-level information (proteomics profiles) is contained in the client's servers and are never shared (dashed lines around clients highlight different physical locations).*"
- 3) We moved the additive secret explanation from panel A, "Federated learning", to the Panel B description and added more details.

Figure 1. FedProt Workflow Overview.

Panel a – Federated workflow overview:

1) Data Preparation: Data owners collect and preprocess MS data, obtain protein intensity and peptide count matrices, and define design matrices before participating as clients. *Personal-level information (proteomics profiles) is contained in the client's servers and are never shared (dashed lines around clients highlight different physical locations).*

2) Federated Learning: Clients communicate with the central server (coordinator) to collaboratively train a global model without revealing their individual datasets, but through the exchange of local model parameters. The clients protect their local parameters using additive secret sharing (blue arrows), *where encrypted parts of masked data are exchanged among clients (blue arrows), ensuring that no single party receives more than one piece of the data from each of the other clients.* In case the data are not numeric, such as protein group names, they are sent to the coordinator without additive secret sharing (green arrows). The coordinator returns updated global parameters to clients (black arrows).

3) Result: After all federated computations, all clients receive the results mathematically equivalent to the results of centralized analysis of pooled dataset with *DEqMS* formatted as a table with abundance fold-changes, confidence intervals, and adjusted p-values.

Panel b: Overview of data communication using SMPC (additive secret sharing) inside FedProt:

The clients protect their local parameters using additive secret sharing. Each client data is masked with a noise mask. The noisy data and the noise masks are splitted into n encrypted parts ($n > 2$). These parts are exchanged among clients (blue arrows) via a relay server, ensuring that no single party receives more than one piece of the data from each of the other clients. After decrypting the received parts, clients sum the data, and send the re-encrypted sums to the coordinator, who decrypts and aggregates the sums to compute the global result. For details see Methods and Supplementary methods.

R2.3 - The sentence, “Due to privacy regulations, finding publicly available multicenter patient-derived data suitable for evaluating FedProt was challenging,” is somewhat ambiguous. It gives the impression that FedProt might not comply with privacy regulations. Perhaps the authors mean that benchmarking cannot be performed on privacy-sensitive data due to the need for data pooling in the non-federated DEqMS. This should be clarified.

We thank the reviewer for highlighting the ambiguity in the original sentence. Indeed, we meant that our study required pooling of the privacy-sensitive data due to the need to perform non-federated centralized DEqMS analysis which results we use as ground truth.

To avoid ambiguity, we have revised the sentence to clarify that the challenge was not about FedProt’s compliance with privacy regulations but rather the difficulty of benchmarking using privacy-sensitive, multicenter patient data. The revised text now reads:

“Due to privacy regulations and data sharing restrictions, finding publicly available multicenter patient-derived data suitable for evaluation, given the need for data pooling in centralized analysis to establish the baseline, was challenging.”

We believe this clarification accurately reflects our intent and addresses the reviewer’s comment.

R2.4 - The manuscript does not explore other proteomics analysis task could be performed under privacy constraints. For example, could federated PCA be used to detect outliers or batch effects? Could these or similar steps be incorporated into FedProt?

Thank you for your suggestion. We agree that incorporating methods such as federated PCA for outlier detection and batch effect analysis could further enhance the capabilities of FedProt.

The FeatureCloud platform is designed to integrate multiple applications into a workflow. It enables various analysis tasks while maintaining privacy. For example, dimensionality reduction methods such as federated singular value decomposition (Hartebrodt, Röttger, and Blumenthal 2024) and clustering techniques like k-means clustering (Hartebrodt 2022) are already available as FeatureCloud apps. These can be combined with FedProt to perform tasks like outlier detection and batch effect analysis. Therefore, incorporating federated PCA or similar methods is certainly possible within the current framework and does not require modifications to the FedProt app itself. However, these applications have not yet been specifically tested on proteomics datasets.

We have addressed this by updating the Methods section to highlight the capability of FeatureCloud to integrate multiple applications into workflows:

“Additionally, the FeatureCloud platform supports the integration of multiple applications into workflows. This allows FedProt to be combined with other privacy-preserving analysis tasks, such as federated singular value decomposition⁵⁴ and clustering techniques like k-means clustering⁵⁵ available as FeatureCloud apps.”

Minors

R2.5 - Lines 64-80 are quite technical and full of abbreviations. Authors could simplify the text for a more general/non-proteomics readership; and stick to most important abbreviations, i.e. DDA vs DIA.

We appreciate the reviewer's suggestion to simplify the text for a broader audience. We have revised the manuscript to enhance clarity and readability without sacrificing the detailed methodological distinctions essential for our study. In particular, we have retained the classifications of label-free quantification (LFQ) and tandem mass tag (TMT) labeling. In proteomics, it is common to differentiate between data acquisition strategies (DDA versus DIA) and quantification methods (LFQ versus labeled), distinctions that are directly relevant to the data presented in our study (LFQ DIA and TMT DDA). Retaining these details is crucial for ensuring both the reproducibility and accurate interpretation of our results.

The revised paragraph:

“Techniques like data-independent acquisition (DIA) mass-spectrometry^{MS} allow simultaneous quantification of thousands of proteins³ with wide proteome coverage and low missing values^{4,5}. By systematically fragmenting all ions within predefined mass ranges, DIA ensures broad ~~DIA systematically fragments all ions within predefined m/z windows, ensuring comprehensive~~ and unbiased peptide ~~quantification and~~ identification⁶. This allows novel peptide identification and provides a deep understanding of protein abundance and post-translational modifications, crucial in clinical proteomics⁷.

In parallel, data-dependent acquisition (DDA), when combined with methods for peptide labeling with tandem mass tags (TMT), ~~MS, accompanied by peptide labeling techniques such as tandem mass tags (TMT)~~, has evolved as a versatile clinical proteomics technique⁸. This ~~approach~~TMT labeling allows simultaneous comparison of peptide abundances across multiple samples in a single MS run ~~and provides the highest accuracy of all relative quantitative proteomic techniques~~. However, ~~it~~but comes with high costs and strict experiment design requirements⁹.

Typically, ~~DIADIA-MS~~ is usually performed without peptide labeling, termed label-free quantification (LFQ). This ~~approach~~DIA-LFQ method ~~iss-are~~ cheaper and requires fewer sample preparation steps, but the accurate quantification of low-abundance proteins is limited¹⁰. Thus, both label-free DIA and DDA with labeling ~~DIA-LFQ and DDA-TMT methods~~ provide unique strengths and are recognized methods in clinical proteomics.”

R2.6 - Line 194 - “We assumed that collaborating parties could agree on using a uniform data preprocessing protocol.” Shouldn't a uniform pre-processing protocol be crucial for such analysis? Maybe a more explicit sentence here would be a better fit? “FedProt requires that all parties agree ... “

We thank the reviewer for this important consideration. While the FedProt method includes normalization methods and considers center-specific batch effects, both can accommodate minor variations in preprocessing. Our results (as detailed in our response to comments R3.1 and R3.3) indicate that integrating non-uniformly preprocessed quantification tables yields performance comparable to conventional meta-analyses in case of comparing with uniformly preprocessed data analyzed centrally (**Fig. S15c**). However, when compared to central analysis on data preprocessed non-uniformly, FedProt shows the same result while the results of meta-analyses diverge from central analysis (**Fig. S15f**). Therefore we conclude that non-uniform preprocessing is a fallback rather than a preferred option. If researchers are

interested in obtaining results as close as possible to the central analysis of pooled data, then performing uniform preprocessing is highly recommended. To clarify in the manuscript, we have updated the text to read:

“We assumed that collaborating parties could agree on using a uniform data preprocessing protocol to ensure optimal data integration. Although FedProt can tolerate minor variations in preprocessing if a strictly uniform protocol is not feasible, our evaluations on balanced dataset (see Supplementary Results section) indicate that results from non-uniform preprocessing are similar to those obtained by centralized DEqMS analysis. Nevertheless, we strongly recommend that all parties use the same quantification software for preprocessing whenever possible.”

R2.7 - Line 459-460 - these statements are unclear. Please consider rewriting.

Thank you for highlighting the lack of clarity in this section. We have revised the sentences. The updated text is as follows:

“(X^i)^T X^i and (X^i)^T Y^i do not reveal any patient-level data and can be shared with the server. To minimize the rare risk of data exposure, clients independently detect and resolve issues during the data validation step (PG detected in only one cohort sample are replaced with NA, Figure 3, step 1).”

Reviewer #3 (Remarks to the Author):

The authors of this study developed FedProt — a tool utilizing federated learning for multi-centered data integration. Proteomics data generated in this study is from five different institutions using different mass spectrometers. Moreover, the pooled data combined with DEqMS analysis results were used as the ground truth for the comparison between different methods. In various comparative analyses, FedProt consistently outperformed other meta-analysis methods in terms of accuracy. The development of FedProt indeed addresses a significant issue in proteomics data analysis. However, the analytical results and comparison methods presented by the authors have certain limitations.

Below are some suggestions:

***R3.1 - 1.** The authors mentioned in lines 194-195, "We assumed that collaborating parties could agree on using a uniform data preprocessing protocol". However, when collecting data from multiple centers, each center may use different protocols. To increase the usability, it would be beneficial to assess the performance of FedProt with data processed using different preprocessing protocols.*

We thank the reviewer for highlighting the practical challenge of integrating data processed with different protocols across centers. In our study, when we refer to a "uniform data preprocessing protocol" (lines 194-195), we mean that collaborating centers agree on using the same quantification software (e.g., MaxQuant/Proteome Discoverer for DDA or MaxQuant-MaxDIA/DIA-NN for DIA). This definition refers solely to the software used for quantification, while downstream steps such as normalization if needed or batch correction are managed by FedProt. We note that while uniformity in raw data quantification is essential for optimal integration, FedProt is specifically designed to handle variability in subsequent processing steps.

We have taken into account differences in sample preparation protocols and LC-MS set-ups. In both datasets used for FedProt evaluation, each center used its own sample preparation methods and LC-MS instrumentation. As recently demonstrated (Distler et al. 2024), particularly for DIA data, differences in LC-MS set-ups still yield highly accurate and precise quantitative measurements across centers. Therefore, in general, consistency between centers in sample preparation or a particular LC-MS instrument is not a requirement for FedProt.

If the concern relates to differences in normalization methods or batch effect correction methods, we would like to clarify that FedProt is designed to handle such variability. For TMT-DDA data, FedProt applies built-in median normalization and internal reference scaling, whereas for DIA data the outputs from MaxDIA are pre-normalized and ready for integration.

If the variability refers to differences in quantification software (e.g., using DIA-NN versus MaxQuant), we have addressed this issue in our analysis, as detailed in our response to comment R3.3. Our additional analysis on a bacterial dataset (see **Fig. S15**) confirmed that when different preprocessing protocols are employed, FedProt's performance remains robust, although the outcomes are comparable to those obtained via conventional meta-analyses.

In cases where reanalysis of raw data is not feasible and different preprocessing protocols have been used, FedProt can still be applied. The main challenge in these situations is the inconsistency in protein groups, which may prevent optimal intersection of datasets across centers; therefore, we recommend collapsing protein groups to gene names to facilitate integration, acknowledging that this approach might result in the loss of isoform-specific information.

In other words, while FedProt is capable of integrating data with heterogeneous preprocessing, such an approach should be considered a fallback option rather than the optimal strategy. To achieve better results, it is necessary to unify not only the data analysis but also raw data quantification software protocols, a task that is beyond the scope of this paper.

In summary, while FedProt is flexible enough to integrate data processed by different tools, we strongly recommend that all centers adopt the same quantification software to ensure optimal data integration. We hope this clarification addresses the reviewer's concerns.

R3.2 - 2. Since FedProt can integrate data from multiple centers, could it also be used to integrate data from different studies? This would be particularly practical for research groups wishing to use data from databases such as PRIDE. It could be extremely beneficial for rare disease studies that may struggle to collect enough samples from a single center. However, as mentioned in point 1, these studies may not use identical preprocessing protocols. I would suggest that the authors present some results demonstrating the effectiveness of FedProt in handling such cases.

We thank the reviewer for highlighting the potential of FedProt to integrate data from different studies. The FedProt method is designed to accommodate variability in sample preparation and raw data acquisition, as demonstrated in our analyses of bacterial and human serum datasets (see responses to R3.1 and R3.3). Thus, FedProt is well-suited to combining datasets from different studies, including those available in public repositories such as PRIDE.

Although optimal integration is achieved when quantification software and preprocessing protocols are uniform, FedProt can still address the challenges posed by heterogeneous preprocessing, provided that the central analysis remains biologically meaningful. In this context, it is important to note that the success of the integration also depends on the consistency of the target class labels and covariates across datasets.

To illustrate this, we analyzed publicly available data for clear-cell renal cell carcinoma (ccRCC) using three datasets: PDC000127 (Clark et al. 2019), PXD042844 (Zhang et al. 2023), PXD030344 (Qu et al. 2022) (see **Supplementary Table S5**). In our analysis, we utilized the published proteomics data matrices, with gene names serving as the common identifier across datasets, and did not use peptide count information. While the PXD042844 and PXD030344 datasets were median normalized and log-transformed, the PDC000127 dataset was used as provided online. We filtered the data to keep only rows with at least one value per target class. The principal component analysis plots along with the intersection of feature (gene) names are shown on **Supplementary Fig. S16**.

To evaluate FedProt and meta-analyses, we performed differential abundance analysis between tumor and control samples. Same data tables, pooled, were centrally analyzed using the DEqMS method to establish the ground truth.

Table S5. Clear-cell renal cell carcinoma dataset description.

	Groups		Set-up
	Control	Tumor	
PDC000127	84	110	Orbitrap Fusion Lumos
PXD042844	114	115	Q Exactive HF-X
PXD030344	232	232	Q Exactive HF-X

Our analysis confirmed that FedProt applied to the ccRCC datasets preprocessed differently again yielded results the same as centralized analyses (**Figure S17, Table S8**). Specifically, the maximum absolute differences between FedProt results and those from the centralized DEqMS analysis are no greater than 8×10^{-13} for both log₂FC and negative log-transformed adjusted p-values. At the same time, however, the best meta-analysis performance in terms of Jaccard similarity coefficient was 0.86. These findings clearly demonstrate that FedProt can effectively analyze data from different studies, even when sample preparation and data preprocessing protocols vary. This underscores the potential utility of FedProt in studies where sample sizes from single centers may be limited, such as understudied, orphan or rare disease studies, particularly when raw data is unavailable and uniform *in silico* preprocessing cannot be performed. We have added the analysis on ccRCC dataset to Supplementary results and to the updated Results section in the main part:

FedProt effectively integrates heterogeneous protein quantification data from multiple centers despite variations in LC–MS setups, software versions, and preprocessing protocols, achieving performance equivalent to centralized DEqMS analysis while outperforming meta-analysis approaches when raw data reanalysis is unfeasible. ...

Furthermore, integration of publicly available proteomics datasets for clear-cell renal cell carcinoma^{40–42} (**Supplementary Table S5**) using gene names as the common identifier further confirmed FedProt's high consistency with centralized DEqMS, as evidenced by negligible differences in Log₂FC and negative log-transformed adjusted p-values (see **Supplementary results**). This evaluation demonstrates FedProt's capability of handling non-uniform preprocessing and its potential utility for multi-center studies, particularly in scenarios where access to raw data is limited.

Figure S16 — Overview of the clear-cell renal cell carcinoma (ccRCC) dataset. Panels a and b: principal component analysis (PCA) plots for data used for differential expression analysis (panel a) and after batch effect correction using limma removeBatchEffect function (panel b); panel c: the number of unique gene names in intensity matrices after filtering (keeping only rows that contain at least one value for each target class).

Figure S17 — The performance of FedProt and meta-analyses compared to results of central DEqMS analysis with ccRCC dataset (combination of 3 studies). Panels a and b: The comparison of negative log-transformed adjusted p-values (panel a) and logFC (panel b) computed by FedProt or meta-analysis methods with the centralized DEqMS analysis. The thin gray line is the diagonal. Panel c: the dependency of the Jaccard similarity coefficient on the number of top-ranked proteins identified by the centralized DEqMS and decentralized approaches, proteins were ranked based on their decreasing negative log-transformed BH-adjusted p-values and not filtered by log2FC.

Table S8. Mean and maximum absolute differences in $|\log_2FC|$, negative log-transformed adjusted p-values, Jaccard similarity coefficients, and error rates (false positives and false negatives) for the results of FedProt and selected meta-analysis approaches compared to centralized DEqMS results using ccRCC dataset (combination of 3 studies). For Jaccard similarity coefficients, FP and FN threshold of $|\log_2FC| > 0.5$ and $\text{adj.p-value} < 0.05$ were used. The best performance is highlighted in bold.

Method	$ \log_2FC $		Adj. p-values		FP	FN	Jaccard similarity coefficient
	Mean difference	Maximal difference	Mean difference	Maximal difference			
FedProt	0.0	0.0	0.0	0.0	0.0	0.0	1.0
Fisher	0.0	0.0	0.0	0.0	0.0	0.0	0.85
Stouffer	0.0	0.0	0.0	0.0	0.0	0.0	0.85
RankProd	0.0	0.0	0.0	0.0	0.0	0.0	0.65
REM	0.0	0.0	0.0	0.0	0.0	0.0	0.3

FedProt	1.08E-15	8.44E-15	6.03E-14	7.11E-13	0	0	1
Fisher	6.08E-02	6.35E-01	9.72E+00	1.63E+02	137	153	0.86
Stouffer	6.08E-02	6.35E-01	7.18E+00	1.18E+02	137	153	0.86
REM	6.11E-02	5.98E-01	2.52E+01	2.57E+02	51	391	0.78
RankProd	6.08E-02	6.35E-01	2.89E+01	2.86E+02	13	462	0.75

R3.3 - 3. Different centers may use different quantification software, such as MaxQuant and Proteome Discoverer for DDA, or MaxDIA and DIA-NN for DIA. These tools can produce different results in protein quantification and identification. Additionally, many centers may only store the quantification tables for long-term storage, rather than the raw data. It would be helpful to demonstrate the performance of FedProt in integrating quantification tables from different software.

We thank the reviewer for raising an important point regarding the use of different quantification software across centers and the potential limitation of having only quantification tables available.

For data requiring normalization, such as DDA-TMT data, FedProt first applies normalization methods to reduce experimental variability (e.g., differences due to TMT-plex effects and sample load). While these procedures effectively address within-center experimental variations, they do not fully correct for discrepancies introduced by the use of different quantification pipelines. For data that are already normalized (for example, LFQ-DIA data processed via MaxLFQ), FedProt directly utilizes the provided values without additional normalization.

To minimize between-center variability, FedProt incorporates center-specific covariates into the linear regression model. This approach accounts for batch effects related to both preprocessing differences and center-specific differences.

Although we strongly recommend using the same software version between centers, we agree that sometimes only quantification tables are available and there is no possibility to re-run quantification on raw data. To evaluate FedProt performance in integrating heterogeneous quantification tables, we used the bacterial dataset that was preprocessed using different quantification software across multiple centers (see **Table S4**). So, not only LC-MS/MS set-up were different between centers, but also raw data preprocessing (quantification software run) was done in different research centers on different machines.

Table S4. Quantification software used for preprocessing bacterial dataset with LC-MS/MS measurement overview. Number of samples in each cohort in each condition. M9 is the medium used to grow *E. coli*. Quantifications were run in different research centers on different machines.

	Groups		Set-up	Quantification software
	M9 Pyruvate	M9 Glucose		
Lab A	10	10	Evosep One – Exploris 480	Spectronaut 17.5
Lab B	10	9	nanoElute – timsTOF Pro	DIA-NN 1.8
Lab C	9	10	Ultimate3000 – Orbitrap Fusion Lumos	DIA-NN 1.8

Lab D	10	10	EASY-nLC 1200 – Exploris 480	DIA-NN 1.8.1
Lab E	10	10	Ultimate3000 – QE-HFX	Spectronaut 17.2

Indeed, different tools and tool's versions used for quantification may lead to differences in protein identification. Those may lead to challenges in obtaining an optimal intersection of protein groups across centers. As shown in **Figure S14**, the number of protein groups available for analysis is not negatively affected by the difference in data preprocessing.

As described in the DIA-NN documentation (Demichev 2025), the intersection of protein groups for DIA data between centers can be improved by “imitating” enabling “match between runs” by first creating a spectral library in one center and then using this library to analyse the other center's data. When raw data reanalysis is not possible, and only quantification tables are available, collapsing protein groups to gene names may help (Adamowicz et al. 2023). In general, FedProt is designed to integrate protein group (PG × sample) matrices across centers regardless of the quantification software used.

Our results using non-uniform raw data quantification in different centers if we compare to centralized DEqMS analysis on uniformly preprocessed data, demonstrate that FedProt can accommodate slight variations, but overall the result in such a case is similar to best meta-analyses performance (see **Fig. S15, panels a-c, and Table S11**). At the same time, if we compare it with the results of centralized DEqMS on non-uniformly quantified data, we can see that FedProt produces the same results as the central analysis with negligibly small deviations (**Fig. S15, panels d-f, and Table S11**). Thus, if raw files are available, it is better to preprocess them as uniformly as possible. If there is no access to raw data, FedProt shows the same result as central analysis on the same data, outperforming meta-analyses.

We hope that this clarification meets the reviewer's concerns. We have added the corresponding section into Supplementary results. Additionally, as we mentioned in reply to **R2.6 comment**, we modified the Result section of the manuscript:

“We assumed that collaborating parties could agree on using a uniform data preprocessing protocol. Although FedProt can tolerate minor variations in preprocessing if a strictly uniform protocol is not feasible, our evaluations on balanced dataset (see Supplementary Results section) indicate that results from non-uniform preprocessing are similar to those obtained by centralized DEqMS analysis. Nevertheless, we strongly recommend that all parties use the same quantification software for preprocessing whenever possible”.

And:

*“FedProt effectively integrates heterogeneous protein quantification data from multiple centers despite variations in LC–MS setups, software versions, and preprocessing protocols, achieving performance equivalent to centralized DEqMS analysis while outperforming meta-analysis approaches when raw data reanalysis is unfeasible. Using a bacterial dataset processed with different quantification software (**Supplementary Table S4**), FedProt delivered results comparable to centralized analysis (see **Supplementary results and Figure S15**)”.*

Figure S14 — The number of unique PGs (panel a) and unique protein IDs (panel b) in quantification software outputs for uniform and non-uniform preprocessing for the bacterial dataset (preprocessed separately for 5 centers, union of all).

Table S11. Jaccard similarity coefficients, and error rates (false positives and false negatives) for the results of FedProt and meta-analysis approaches for the bacterial dataset with non-uniform compared to centralized DEqMS results for the bacterial dataset with uniform and non-uniform (for FP and FN, threshold of $|\log_{2}FC| > 0.5$ and $adj.p\text{-value} < 0.05$ were used). The best performance is highlighted in bold.

Method	compared to uniform, centralized DEqMS			compared to non-uniform, centralized DEqMS		
	FP	FN	Jaccard similarity coefficient	FP	FN	Jaccard similarity coefficient
FedProt	94	10	0.857	0	0	1
Fisher	95	10	0.855	3	2	0.993
Stouffer	95	10	0.855	3	2	0.993
REM	95	13	0.851	14	16	0.959
RankProd	22	124	0.776	0	186	0.74

Figure S15 — The performance of FedProt and meta-analyses using non-uniformly preprocessed data compared to results of central DEqMS analysis with uniformly preprocessed data (panels a-c) and non-uniformly preprocessed data (panels d-f). Panels a, b and d, e: The comparison

of negative log-transformed adjusted p-values (panels a, d) and logFC (panels b, e) computed by FedProt or meta-analysis methods (y-axis, non-uniform preprocessing) with the centralized DEqMS analysis (x-axis, uniform preprocessing). The thin gray line is the diagonal. Panels c and f: The dependency of the Jaccard similarity coefficient on the number of top-ranked proteins identified by the centralized DEqMS and decentralized approaches, proteins were ranked based on their decreasing negative log-transformed BH-adjusted p-values and not filtered by log₂FC.

R3.4 - 4. Since statistical significance is based on the adjusted p-value, it would be interesting to investigate the results using different cutoffs for log2 fold-change. The authors set different cutoffs for various analyses and datasets, such as human serum, bacterial data, and data imbalance comparisons. Is it possible that different meta-analysis methods may show varying accuracy based on different log2 fold-change cutoffs?

In practical applications, the selection of logFC and adj. p-value thresholds depend on many factors including the experimental system, specific phenotype, and research objectives. Although in many studies researchers choose cutoffs $|\logFC|>2$ or $|\logFC|>1$, in the human serum dataset with a small effect size for proteins, setting a logFC cutoff of 1 resulted in no detected true positives, meaning that no differentially abundant proteins were identified even in the centralized scenario. In the bacterial dataset, only 2.64% of all measured proteins were significantly (adj.p-value<0.05) differentially abundant with $|\logFC|>1$ (**Supplementary Figure S6**). Therefore, for the purposes of benchmark we reduced the logFC thresholds to 0.25 and 0.5 for human serum and bacterial datasets, respectively. These settings yielded in around 10% of differentially abundant protein groups in both datasets.

Figure S6.

The proportion of differentially abundant proteins in (a) bacterial dataset and (b) human serum dataset in centralized analysis (DEqMS) under different adj.p-value and logFC cutoffs. The red square marks selected cutoff for each dataset.

To evaluate the impact of varying logFC and adj. p-value cutoffs on the performance of FedProt and meta-analysis methods, we calculated Jaccard indices and numbers of FP and FN for a range of logFC and adj. p-value cutoffs (**Supplementary Figures S7 and S8**).

In both datasets, regardless of the chosen thresholds, FedProt results always remained the same as the results of centralized analyses. Meta-analyses results, in contrast, tended to diverge more from the results of centralized analyses when logFC threshold was relaxed. Varying p-value threshold between 0.001 and 0.05 had little influence on the list of differentially abundant proteins and therefore in most cases only slightly affected performance estimates.

To illustrate this observation, we have added a new **Supplementary Results section (Supplementary Figures S6-S8)** and also expanded the Results section to include a discussion of these findings:

*“In addition, we evaluated a range of logFC and adj.p-value thresholds to examine how different cutoffs affect FedProt and meta-analysis methods performance (see **Supplementary Results** section). ... FedProt maintained robust performance even under varying thresholds, although the meta-analyses showed greater discrepancies, especially for smaller logFC cutoffs, as for the human serum dataset (**Supplementary Fig. S6-S8**).”*

We believe these additional analyses address the reviewer’s concerns and further demonstrate the robustness of our federated approach under varying threshold conditions.

Jaccard Index for Bacterial dataset

Jaccard Index for Human serum dataset

Figure S7.

Consistency between centralized and decentralized methods for the bacterial and human serum datasets in terms of Jaccard index for different adj.p-value and logFC cutoffs. The red square marks the logFC and adj.p-value cutoff selected for the main analysis.

Figure S8. Error rates (false positives – panels a, b; and false negatives – panels c, d) for the results of FedProt and selected meta-analysis approaches compared to centralized DEqMS results using bacterial (panels a, c) and human serum (panels b, d) datasets for different adj.p-value and logFC cutoffs. The red square marks the logFC and adj.p-value cutoff for the main analysis.

R3.5 - 5. It would be very helpful for readers to understand the workflow, including normalization, filtering, and data integration, by using a simple example or figure like Figure S11.

We thank the reviewer for the suggestion to enhance the clarity of our workflow. In response, we have revised Figure 3 to include additional details illustrating the validation, filtering, and normalization steps. The updated figure:

Figure 3. Scheme of the FedProt workflow.

Steps that involve federated computations are shown in green. The corresponding stages of DEqMS workflow are shown on the right. Median normalization from the PRONE⁴⁸ R package (<https://github.com/daisybio/PRONE>) was used. The validation, filtering, normalization and design mask creation steps involving 3 clients (C1, C2, C3) are shown on the left. PG denotes protein groups, j are sample numbers, A and B are target classes that are compared during differential abundance analysis. On step 1, the PG3 value for sample j5 is replaced with NA because there is only one not-NA value for this PG in the client data. On step 2, the PG5 value for sample j5 is replaced with NA because there is only one not-NA value for this PG in the target class for this client. After that, the whole PG5 group is removed from all clients because of too few non-missing values (less than f=0.75 in this case). On the design mask creation step, client 3 for PG1 is excluded because it has no data, same for PG4 of client 2; and, because PG3 is missing in client 1, the client 3 became the new reference client and also excluded from computation (see **Supplementary methods** for more details).

* — The normalization step is optional and could be turned off by the coordinator. Because the data derived from a DIA LFQ experiment and processed through MaxLFQ protein quantification⁴⁵ are usually already normalized, so no additional preprocessing is needed except for filtering by protein group Q-value. In case of TMT data, ~~also no additional preparation is needed, except for filtering out decoys, contaminant and reverse protein groups are required.~~ The normalization by median across all centers and internal reference scaling inside each center can be performed during FedProt run. We suggest enabling the “match between runs” option during raw MS data quantification.

** — The step is for the federated approach only.

R3.6 - 6. To help readers understand the differences between FedProt and other meta-analysis methods, it would be helpful to generate a figure comparing the methods, similar to Flimma (Fig 1, <https://genomebiology.biomedcentral.com/articles/10.1186/s13059-021-02553-2>).

We thank the reviewer for this suggestion and agree that a visual abstract would help readers to better understand the differences between FedProt, central analysis and meta-analysis methods. However, due to the display limit (up to 6 items, figures and/or tables) we have decided to exclude the Flimma-like figure from the first version of the manuscript as we did not find a way to combine it with any other figure. On your recommendation, we have now added this figure to the supplement (**Figure S2**) (“An illustration of the workflows is given in Supplementary Fig. S2.”).

Figure S2. Protein abundance analysis workflows in case of multi-center studies. Dashed areas highlight different physical locations.

R3.7 - 7. Is DIA-NN run using a “library-free search” or a “DDA spectra library-dependent” approach? Please address this in the manuscript, as it may influence the accuracy of protein identification and quantification.

We thank the reviewer for highlighting the importance of clarifying the DIA-NN analysis approach used in our study. Specifically, we provided a FASTA file containing the relevant protein sequences, allowing DIA-NN to generate the spectral library *in silico* during the analysis. We did not use a pre-existing spectral library.

We have updated the manuscript to reflect this more clearly:

“An *in silico* spectral library was generated in each DIA-NN run from *E. coli* MG1655 (taxID 83333) protein sequence database (Uniprot UP000000625, 4448 entries) **provided via a FASTA file (no pre-existing spectral library was utilized).**”

R3.8 - 8. *I suggest the authors carefully reconsider the naming conventions for the tables and figures. For example, the column names in Table S2 are quite confusing. From my understanding, B1, B2, and B3 should refer to Batch 1, 2, and 3, not condition B.*

Thank you for your suggestion. We acknowledge that the original naming in Table S2 may lead to ambiguity. Specifically, the labels B1, B2, and B3 are intended to denote the three cohorts (or batches), while the label "B" was used for condition B. To eliminate any potential confusion, we have revised the cohort labels to C1, C2, and C3 (as **C**lients), and retained the condition names as "A" and "B".

R3.9 - 9. *In Figure 2a and c, the y-axis is labeled "other methods," which feels a bit odd. Would it be better to change it to "FedProt and other meta-analysis methods"?*

We agree with your suggestion. We have updated the y-axis label in Figures 2a and 2c together with supplementary figures, to "FedProt and other meta-analysis methods".

R3.10 - 10. *In Figure 2b, the labels on the x-axis are not capitalized. It would be better to make them consistent with Figure 2D.*

Thank you for pointing out the inconsistency in the x-axis labels of Figure 2b. We have now capitalized these labels to ensure consistency with Figure 2d.

R3.11 - 11. *Line 246. "Tables 2" should be "Table 2".*

Thank you for spotting this mistake. We have corrected it in the revised manuscript.

Reviewer #3 (Remarks on code availability):

R3.12 - 1. *The code is well-written, but the documentation regarding Docker is unclear. It may need to run Docker when executing FedProt, but this is not mentioned. Users who are not familiar with Docker may encounter difficulties in executing or installing the program.*

We thank the reviewer for the valuable comment regarding the documentation. In response, we have updated the README to more clearly state the requirements for installing and running FedProt (<https://github.com/Freddsle/FedProt?tab=readme-ov-file#running-the-app>). In particular, we now emphasize that although Docker must be installed and functioning on the user's system, there is no need to manually run any Docker command.

R3.13 - 2. *Having a log file to store the executed commands and the process will be a great help for users to review or troubleshoot issues.*

We appreciate the reviewer's comment. In FedProt's implementation, a log file capturing all processing steps is indeed available to users. The logs are stored in a log folder within the directory where the controller is initiated. Additionally, in the GUI, the log file can be easily downloaded through a "Logs" button. Furthermore, if FedProt is operating in the FeatureCloud test environment, users can view the log files directly in the testing section. We have updated the README to describe these details.

R3.14 - 3. *In "Prerequisite" section of README, There is a typo in the comment of the second command - # first, create and go "the" the dir, where test folder will be created. It should be - # first, create and go "to" the dir, where test folder will be created.*

We thank the reviewer for catching this and have corrected the README.

References

- Adamowicz, Klaudia, Lis Arend, Andreas Maier, Johannes R. Schmidt, Bernhard Kuster, Olga Tsoy, Olga Zolotareva, Jan Baumbach, and Tanja Laske. 2023. "Proteomic Meta-Study Harmonization, Mechanotyping and Drug Repurposing Candidate Prediction with ProHarMeD." *Npj Systems Biology and Applications* 9 (1): 49.
- Clark, David J., Saravana M. Dhanasekaran, Francesca Petralia, Jianbo Pan, Xiaoyu Song, Yingwei Hu, Felipe da Veiga Leprevost, et al. 2019. "Integrated Proteogenomic Characterization of Clear Cell Renal Cell Carcinoma." *Cell* 179 (4): 964–83.e31.
- Demichev, Vadim. 2025. *DiaNN: DIA-NN - a Universal Automated Software Suite for DIA Proteomics Data Analysis*. Github. <https://github.com/vdemichev/DiaNN>.
- Distler, Ute, Han Yoo, Oliver Kardell, Dana Hein, Malte Sielaff, Christian Leps, Marian Scherer, et al. 2024. "Multicenter Evaluation of Label-Free Quantification in Human Plasma: Benchmarking with a High Dynamic Range Multispecies Sample Set." *Research Square*. <https://doi.org/10.21203/rs.3.rs-5618718/v1>.
- Eldjarn, Grimur Hjorleifsson, Egil Ferkingstad, Sigrun H. Lund, Hannes Helgason, Olafur Th Magnusson, Kristbjorg Gunnarsdottir, Thorunn A. Olafsdottir, et al. 2023. "Large-Scale Plasma Proteomics Comparisons through Genetics and Disease Associations." *Nature* 622 (7982): 348–58.
- Hartebrodt, Anne. 2022. "Federated Unsupervised Machine Learning." Syddansk Universitet. Det Naturvidenskabelige Fakultet. <https://doi.org/10.21996/Z4YW-JM67>.
- Hartebrodt, Anne, Richard Röttger, and David B. Blumenthal. 2024. "Federated Singular Value Decomposition for High-Dimensional Data." *Data Mining and Knowledge Discovery* 38 (3): 938–75.
- Karr, Alan F., Xiaodong Lin, Ashish P. Sanil, and Jerome P. Reiter. 2005. "Secure Regression on Distributed Databases." *Journal of Computational and Graphical Statistics: A Joint Publication of American Statistical Association, Institute of Mathematical Statistics, Interface Foundation of North America* 14 (2): 263–79.
- Matschinske, Julian, Julian Späth, Mohammad Bakhtiari, Niklas Probul, Mohammad Mahdi Kazemi Majdabadi, Reza Nasirigerdeh, Reihaneh Torkzadehmahani, et al. 2023. "The FeatureCloud Platform for Federated Learning in Biomedicine: Unified Approach." *Journal of Medical Internet Research* 25 (July):e42621.
- Polanin, Joshua R. 2018. "Efforts to Retrieve Individual Participant Data Sets for Use in a Meta-Analysis Result in Moderate Data Sharing but Many Data Sets Remain Missing." *Journal of Clinical Epidemiology* 98 (June):157–59.
- Qu, Yuanyuan, Jinwen Feng, Xiaohui Wu, Lin Bai, Wenhao Xu, Lingli Zhu, Yang Liu, et al. 2022. "A Proteogenomic Analysis of Clear Cell Renal Cell Carcinoma in a Chinese Population." *Nature Communications* 13 (1): 2052.
- Speich, Benjamin, Dmitry Gryaznov, Jason W. Busse, Viktoria L. Gloy, Szimonetta Lohner, Katharina Klatte, Ala Taji Heravi, et al. 2022. "Nonregistration, Discontinuation, and Nonpublication of Randomized Trials: A Repeated Metaresearch Analysis." *PLoS Medicine* 19 (4): e1003980.
- Zhang, Hailiang, Lin Bai, Xin-Qiang Wu, Xi Tian, Jinwen Feng, Xiaohui Wu, Guo-Hai Shi, et al. 2023. "Proteogenomics of Clear Cell Renal Cell Carcinoma Response to Tyrosine Kinase Inhibitor." *Nature Communications* 14 (1): 4274.